# FEDHPO-BENCH: A BENCHMARK SUITE FOR FEDERATED HYPERPARAMETER OPTIMIZATION

## ABSTRACT

Hyperparameter optimization (HPO) is crucial for machine learning algorithms to achieve satisfactory performance. Its research progress has been boosted by existing HPO benchmarks. Nonetheless, existing efforts in benchmarking all focus on HPO for traditional centralized learning while ignoring federated learning (FL), a promising paradigm for collaboratively learning models from dispersed data. In this paper, we first identify some uniqueness of HPO for FL algorithms from various aspects. Due to this uniqueness, existing HPO benchmarks no longer satisfy the need to compare HPO methods in the FL setting. To facilitate the research of HPO in the FL setting, we propose and implement a benchmark suite FEDHPO-BENCH that incorporates comprehensive FedHPO problems, enables flexible customization of the function evaluations, and eases continuing extensions. We also conduct extensive experiments based on FEDHPO-BENCH to provide the community with more insights into FedHPO. We open-sourced FEDHPO-BENCH at https://github.com/FedHPO-Bench/FedHPO-Bench-ICLR23

## 1 INTRODUCTION

Most machine learning (ML) algorithms expose many design choices, which can drastically impact the ultimate performance. Hyperparameter optimization (HPO) (Feurer & Hutter, 2019) aims at making the right choices without human intervention. To this end, HPO methods usually attempt to solve $\min_{\lambda \in \Lambda_1 \times \cdots \times \Lambda_K} f(\lambda)$, where each $\Lambda_k$ corresponds to the candidate choices of a specific hyperparameter, e.g., taking the learning rate from $\Lambda_1 = [0.01, 1.0]$ and the batch size from $\Lambda_2 = \{16, 32, 64\}$. For each specified $\lambda$, $f(\lambda)$ is the output result (e.g., validation loss) of executing the considered algorithm configured by $\lambda$. A solution $\lambda^*$ found for such a problem is expected to make the considered algorithm lead to superior generalization performance. Research in this line has been facilitated by HPO benchmarks (Gijsbers et al., 2019; Eggensperger et al., 2021; Pineda-Arango et al., 2021), which prepare many HPO problems so that different HPO methods can be effortlessly compared, encouraging fair, reliable, and reproducible empirical studies.

However, existing HPO benchmarks all focus on traditional learning paradigms, where the functions to be optimized correspond to centralized learning tasks. Federated learning (FL) (McMahan et al., 2017; Li et al., 2020a), as a privacy-preserving paradigm for collaboratively learning a model from distributed data, has not been considered. Actually, along with the increasing privacy concerns from the whole society, FL has been gaining more attention from academia and industry. Meanwhile, HPO for FL algorithms (denoted by FedHPO from now on) is identified as a critical and promising open problem in FL (Kairouz et al., 2019).

In this paper, we first elaborate on several differences between FedHPO and traditional HPO (see Section 2.2), which essentially come from FL's distributed nature and the heterogeneity among FL's participants. These differences make existing HPO benchmarks inappropriate for studying FedHPO and, in particular, unusable for comparing FedHPO methods. Consequently, several recently proposed FedHPO methods (Zhou et al., 2021; Dai et al., 2020; Khodak et al., 2021; Zhang et al., 2021; Guo et al., 2022) are evaluated on respective problems and have not been uniformly implemented in one FL framework and well benchmarked.

Motivated by FedHPO's uniqueness and the successes of existing HPO benchmarks, we propose and implement FEDHPO-BENCH, a dedicated benchmark suite, to facilitate the research and application of FedHPO. FEDHPO-BENCH is featured by satisfying the desiderata as follows:

**Comprehensiveness**. FL tasks are diverse in terms of domain, model architecture, heterogeneity among participants, etc. The objective functions of their corresponding FedHPO problems are thus likely to be diverse. Hence, FEDHPO-BENCH provides a comprehensive collection of FedHPO problems for drawing an unbiased conclusion from comparisons of HPO methods.

**Flexibility.** Users may have different levels of privacy and fairness concerns, which may correspond to different multi-objective optimization problems. Meanwhile, the execution time for function evaluation depends on the system condition. Thus, FEDHPO-BENCH allows users to flexibly tailor the FedHPO problems to their privacy protection needs, fairness demands, and system conditions.

**Extensibility.** As a developing field, new FedHPO problems and novel FedHPO methods constantly emerge, and FL's best practice continuously evolves. Thus, we build FEDHPO-BENCH on a popular FL framework, FederatedScope (FS) (Xie et al., 2022), and make it more of a benchmarking tool that can effortlessly incorporate novel ingredients.

To our knowledge, FEDHPO-BENCH is the first FedHPO benchmark. We conduct extensive empirical studies with it to validate its usability and attain more insights into FedHPO.

## 2 BACKGROUND AND MOTIVATIONS

We first give a brief introduction to the settings of HPO and its related benchmarks. Then we present and explain the uniqueness of FedHPO to show the demand for dedicated FedHPO benchmarks. Due to the space limitation, more discussions about related works are deferred to Appendix B.

### 2.1 PROBLEM SETTINGS AND EXISTING BENCHMARKS

In the literature (Feurer & Hutter, 2019), HPO is often formulated as solving $\min_{\lambda \in \Lambda_1 \times \cdots \times \Lambda_K} f(\lambda)$, where each $\Lambda_k$ corresponds to candidate choices of a specific hyperparameter, and their Cartesian product (denoted by $\times$) constitute the search space. In practice, such $\Lambda_k$ is often bounded and can be continuous (e.g., an interval of real numbers) or discrete (e.g., a set of categories/integers). Each function evaluation at a specified hyperparameter configuration $\lambda$ means to execute the corresponding algorithm accordingly and return the value of considered metric (e.g., validation loss) as the result $f(\lambda)$. HPO methods generally solve such a problem with a series of function evaluations. As a full-fidelity function evaluation is extremely costly, multi-fidelity methods exploit low-fidelity function evaluation, e.g., training for fewer epochs (Swersky et al., 2014; Domhan et al., 2015) or on a subset of data (Klein et al., 2017; Petrak, 2000; Swersky et al., 2013), to approximate the exact result. Thus, it would be convenient to treat $f$ as $f(\lambda, b), \lambda \in \Lambda_1 \times \cdots \times \Lambda_K, b \in \mathcal{B}_1 \times \cdots \times \mathcal{B}_L$, where each $\mathcal{B}_l$ corresponds to the possible choices of a specific fidelity dimension.

HPO benchmarks (Gijsbers et al., 2019; Eggensperger et al., 2021; Pineda-Arango et al., 2021) have prepared many HPO problems, i.e., various kinds of objective functions, for comparing HPO methods. To evaluate these functions, HPO benchmarks, e.g., HPOBench (Eggensperger et al., 2021), often provide three modes: (1) "Raw" means truly executing the corresponding algorithm; (2) "Tabular" means querying a lookup table, where each entry corresponds to a specific $f(\lambda, b)$; (3) "Surrogate" means querying a surrogate model that might be trained on the tabular data.

### 2.2 UNIQUENESS OF FEDERATED HYPERPARAMETER OPTIMIZATION

Generally, traditional HPO methods are applicable to FedHPO problems[1], where, in each trial, the value $f(\lambda, b)$ is evaluated, that is to say, an accordingly configured FL training course is conducted, as the dashed black box in Figure 1 illustrates. Conceptually, there are $N$ clients, each of which has its specific data, and a server coordinates them to learn a model $\theta$ collaboratively by an FL algorithm such as FedAvg (McMahan et al., 2017) and FedOpt (Asad et al., 2020). Such FL algorithms are iterative. In the $t$-th round, the server broadcasts the global model $\theta^{(t)}$ to sampled clients; then, these clients make local updates and send the updates back; finally, the server aggregates the updates to produce $\theta^{(t+1)}$. After executing the FL algorithm configured by $\lambda$ for several such rounds, e.g., #round$= T$ according to the specified fidelity $b$, the performance, e.g., best validation loss ever achieved during this FL course, is returned as $f(\lambda, b)$.

---

[1]Despite the various scenarios in literature, we restrict our discussion about FedHPO to one of the most general FL scenarios that have been adopted in existing FedHPO works (Khodak et al., 2021; Zhang et al., 2021).

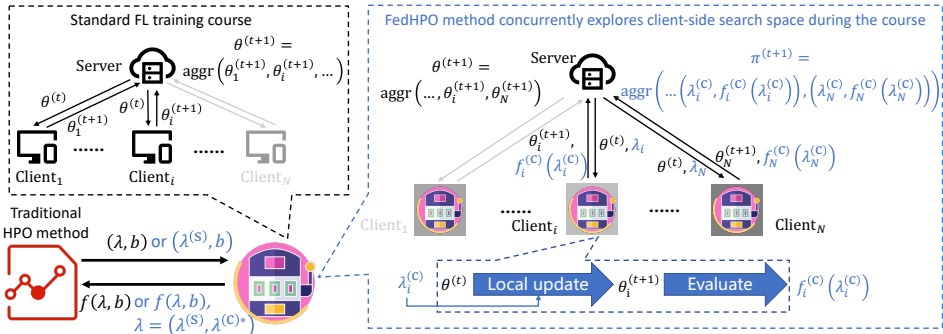

**Figure 1:** Solving a FedHPO problem by a traditional HPO method solely or as the wrapper of a FedHPO method: the $t$-th round of a trial is shown in the dashed black box and dashed blue box, respectively. The faded clients are not sampled in that round; "aggr" denotes a certain aggregation operation; here FedEx is considered as the wrapped FedHPO method, which learns a policy $\pi$ to determine $\lambda^{(c)*}$; each $f_i^{(c)}(\cdot)$ is regarded as a client-specific approximation of $f(\lambda^{(s)}, \cdot)$.

As a distributed ML scenario, the procedure of each FL training round consists of two subroutines—local updates and aggregation. Thus, $\lambda$ can be divided into server-side and client-side hyperparameters according to which subroutine each hyperparameter influences. Denoting it as $\lambda = (\lambda^{(s)}, \lambda^{(c)})$, the original optimization problem can be straightforwardly restated as its bi-level counterpart $\min_{\lambda^{(s)}} f(\lambda^{(s)}, \lambda^{(c)*})$, s.t., $\lambda^{(c)*} = \min_{\lambda^{(c)}} f(\lambda^{(s)}, \lambda^{(c)})$. With such a point of view, let's see how some FedHPO methods leverage the distributed nature of FL to solve the lower-level sub-problem efficiently and approximately.

**Concurrent exploration**. In each FL training course, all sampled clients have executed local updates configured by client-side hyperparameters for one or more rounds. Given a specific $\lambda^{(s)}$, if we regard clients as replicas of the black-box function $f(\lambda^{(s)}, \cdot)$ or, at least, similar such functions whose evaluation results help fitting $f(\lambda^{(s)}, \cdot)$, it is natural to try out client-side hyperparameter configurations client-wisely so that we evaluate $f(\lambda^{(s)}, \lambda^{(c)})$ for more than one $\lambda^{(c)}$s in each FL training course. We summarize this idea as concurrent exploration, which recently proposed FedHPO methods, such as FedEx (Khodak et al., 2021) and FTS (Dai et al., 2020), have instantiated. Specifically, FedEx incorporates concurrent exploration with the weight-sharing trick (Liu et al., 2019) to achieve one-shot learning (i.e., by one FL training course) of a policy $\pi$ for determining the optimal lower-level response $\lambda^{(c)*} = \min_{\lambda^{(c)}} f(\lambda^{(s)}, \lambda^{(c)})$. Hence, we could solve the bi-level optimization problem by letting a traditional HPO method, as a wrapper, choose $\lambda^{(s)2}$ and replace each standard FL training course with a FedHPO method, as the dashed blue box in Figure 1 shows.

As FedHPO methods are fused with the training course, existing HPO benchmarks become unusable for comparing them. Moreover, since this fusion makes the implementations of such FedHPO methods tightly coupled with that of FL training courses, and no existing FL framework has incorporated such FedHPO methods, researchers cannot compare them in a unified way. How we incorporate several recent FedHPO methods into FEDHPO-BENCH and make it extensible is discussed in Section 3.3, and whether concurrent exploration is useful is empirically answered in Section 4.2.

**Personalization**. The non-IIDness among clients' data is likely to make them have different optimal configurations (Koskela & Honkela, 2020), where making decisions by the same policy, such as FedEx, would become unsatisfactory. This phenomenon tends to become severer when federated hetero-task learning (Yao et al., 2022) is considered. Trivially solving the personalized FedHPO problem $\min_{\lambda^{(s)}, \lambda_1^{(c)}, \ldots, \lambda_N^{(c)}} f(\lambda^{(s)}, \lambda_1^{(c)}, \ldots, \lambda_N^{(c)})$, where $\lambda_i^{(c)}$ denotes the client-side hyperparameter configuration of the $i$-th client, is intractable, as the search space exponentially increases with $N$. To promote studying personalized FedHPO, we provide a personalized FedHPO problem featured by heterogeneous tasks among the clients and the corresponding tabular benchmark (see H.1 for a detailed description).

---

[2] In practice, the traditional HPO method also determines the FedHPO algorithm's hyperparameters and a subset of the original client-side search space to be explored. We omit these to keep brevity.

**Multi-objective optimization**. Despite the model's performance, researchers are often concerned about other issues, such as privacy protection and fairness. Regarding privacy, the FL algorithm is often incorporated with privacy protection techniques such as differential privacy (DP) (Kairouz et al., 2019), where the DP algorithm also exposes its hyperparameters. Intuitively, a low privacy budget specified for such algorithms indicates a lower risk of privacy leakage yet a more significant degradation of the model's performance. As for fairness, namely, the uniformity of the model's performances across the clients, more and more FL algorithms have taken it into account (Li et al., 2021a; Wang et al., 2021b), which contains some hyperparameter(s) concerning fairness. Therefore, researchers may be interested in searching for a hyperparameter configuration that guarantees an acceptable privacy leakage risk (e.g., measured by Rényi-DP (Mironov, 2017)) and fairness measurement (e.g., the standard deviation of client-wise performances) while optimizing the model's performance.

Thus, a FedHPO benchmark is expected to expose a vector-valued objective function instead of a scalar-valued one, where the entries of a returned vector could be the quantitative measures corresponding to performance, privacy leakage risk, fairness, etc. Then, researchers are allowed to study multi-objective HPO (Hernández et al., 2021; Deb et al., 2002; Abdolshah et al., 2019).

**Runtime estimation and system-dependent trade-offs**. For the research purpose, an FL training course is usually simulated in a single computer rather than executed in a distributed system. As a result, simply recording the consumed time is meaningless for measuring the cost of a function evaluation in studying FedHPO. Meanwhile, FL's distributed nature introduces a new fidelity dimension—*client_sample_rate*, which determines the fraction of clients sampled in each round. When considering a *client_sample_rate* less than that of a full-fidelity function evaluation, each round of the FL course would take less time because there is less likely to be a straggler. However, *client_sample_rate* correlates with another fidelity dimension—*#round*, where a lower *client_sample_rate* often leads to federated aggregation with larger variance, which is believed to need more rounds for convergence. How we should balance these two fidelity dimensions to achieve more economical accuracy-efficiency trade-offs strongly depends on the system condition, e.g., choosing large *#round* but small *client_sample_rate* when the straggler issue is severe.

As existing HPO benchmarks focus on centralized learning tasks, they overlook a runtime estimation functionality for studying FedHPO. In Section 3.2, we present FEDHPO-BENCH's system model, with which we conduct an empirical study to show the effect of balancing *client_sample_rate* and *#round* in Appendix G.

Due to the uniqueness mentioned above, existing HPO benchmarks are inappropriate for studying FedHPO. FedHPO calls for dedicated benchmarks that incorporate objective functions corresponding to FL algorithms and respecting realistic FL settings.

## 3 OUR PROPOSED BENCHMARK SUITE: FEDHPO-BENCH

We present an overview of FEDHPO-BENCH in Figure 2. Conceptually, FEDHPO-BENCH encapsulates function evaluation and provides a unified interface for HPO methods to interplay with it. Following the design of HPOBench (Eggensperger et al., 2021), function evaluations can be conducted in either of the three modes: *raw*, *tabular*, or *surrogate*. For the raw mode, we chose to build FEDHPO-BENCH upon the well-known FL platform FederatedScope (FS) (Xie et al., 2022), which has provided its docker images so that we can containerize FEDHPO-BENCH effortlessly by executing each FL algorithm in an FS docker container. To generate the lookup table for tabular mode, we truly execute the corresponding FL algorithms with the grids of search space as their configurations. These lookup tables are adopted as training data for the surrogate models, which are expected to approximate the objective functions (more details about this approximation are discussed in Appendix H.4.2). It's important to note that the distributed nature of FL makes it very expensive to run an FL course, so, in FedHPO, the tabular and surrogate modes are much in demand to meet the efficiency requirement. For the convenience of users, we keep FEDHPO-BENCH's interface basically the same as HPOBench's. Meanwhile, we expose extra arguments for users to customize the instantiation of a benchmark. We defer the discussion of the relationship between FEDHPO-BENCH and HPOBench to Appendix B.1.

In this section, we elaborate on three highlights of FEDHPO-BENCH. In addition to the off-the-shelf FL-related components that FS already provides, we contribute new datasets, models, and algorithms

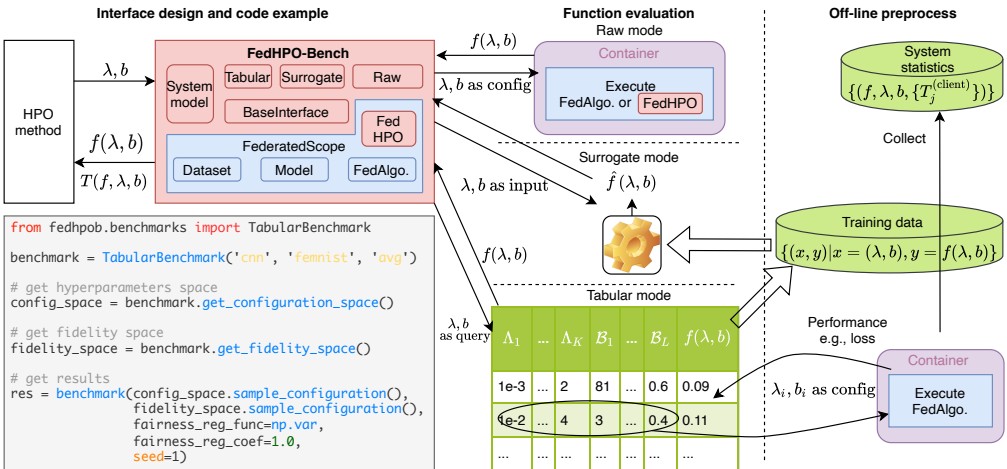

**Figure 2:** Overview of FEDHPO-BENCH.

to FS to prepare a comprehensive collection of FedHPO problems (see Section 3.1). Moreover, users are allowed to flexibly tailor these problems to their specific cases (see Section 3.2). Also, FS's event-driven framework allows us to easily extend FEDHPO-BENCH by incorporating more FedHPO problems and methods, which is valuable for this nascent research direction (see Section 3.3).

## 3.1 COMPREHENSIVENESS

There is no universally best HPO method (Gijsbers et al., 2019). Therefore, for the purpose of fairly comparing HPO methods, it is necessary to compare them on a variety of HPO problems that correspond to diverse objective functions and thus can comprehensively assess their performances.

To satisfy this need, we leverage FS to prepare various FL tasks, where their considered datasets and model architectures are quite different. Specifically, the data can be images, sentences, graphs, or tabular data. Some datasets are provided by existing FL benchmarks, which are readily distributed and thus conform to the FL setting. Some are centralized initially, which we partition by FS's splitters to construct their FL version with various kinds of Non-IIDness among clients. All these datasets are publicly available and can be downloaded and preprocessed by our prepared scripts. More details of these datasets can be found in Appendix D. Then the corresponding suitable neural network model is applied to handle each dataset, involving fully-connected networks, convolutional networks, the latest attention-based model, etc. It is worth noticing that our prepared FL tasks cover both cross-silo and cross-device scenarios. In cross-device scenario, there are a lot more clients and a much lower *client_sample_rate* than in cross-silo scenario.

For each such FL task, we basically employ two FL algorithms, FedAvg and FedOpt, to learn the model, respectively. Then the FedHPO problem is defined as optimizing the design choices of the FL algorithm on each specific FL task. So, without further explanation, we use the triple <dataset, model, algorithm> to index a particular benchmark in the remainder of this paper. We summarize our currently provided FedHPO problems in Table 1, and more details can be found in Appendix H. For each problem, *#round* and *client_sample_rate* are adopted as the fidelity dimensions.

We study the empirical cumulative distribution function (ECDF) for each model type in the cross-silo benchmarks. Specifically, in creating the lookup table for tabular mode, we have conducted function evaluations for the hyperparameter configurations located on a very dense grid over the search space, resulting in a finite set $\{(\lambda, f(\lambda))\}$ for each benchmark. Then we normalize the performances (i.e., $f(\lambda)$) and show their ECDF in Figure 3, where these curves exhibit different shapes. For example, the amounts of top-tier configurations for GNN on PubMed are remarkably less than on other graph datasets, which might imply a less smoothed landscape and difficulty in seeking the optimal configuration. As the varying shapes of ECDF curves have been regarded as an indicator of the diversity of benchmarks (Eggensperger et al., 2021), we can conclude from Figure 3 that FEDHPO-BENCH enables evaluating HPO methods comprehensively.

**Table 1:** Summary of benchmarks in current FEDHPO-BENCH: MF refers to matrix factorization. Rec. and Algo. are short for recommendation and algorithm, respectively. #Cont. and #Disc. denote the number of hyperparameter dimensions corresponding to continuous and discrete candidate choices, respectively. The unit of the budget is either day (d) or second (s).

| Scenario | Model | #Dataset | Domain | #Client | #Algo. | #Cont. | #Disc. | Budget |
|---|---|---|---|---|---|---|---|---|
| Cross-Silo | CNN | 2 | CV | 200 | 2 | 4 | 2 | 20d |
| | BERT | 2 | NLP | 5 | 2 | 4 | 2 | 20d |
| | GNN | 3 | Graph | 5 | 2 | 4 | 1 | 1d |
| | GNN | 1 | Hetero | 5 | 1 | 1 | 1 | 1d |
| | LR | 7 | Tabular | 5 | 2 | 3 | 1 | 21,600s |
| | MLP | 7 | Tabular | 5 | 2 | 4 | 3 | 43,200s |
| Cross-Device | MF | 1 | Rec. | 480,189 | 2 | 3 | 1 | - |
| | LR | 1 | NLP | ∼3300 | 2 | 3 | 1 | 1d |

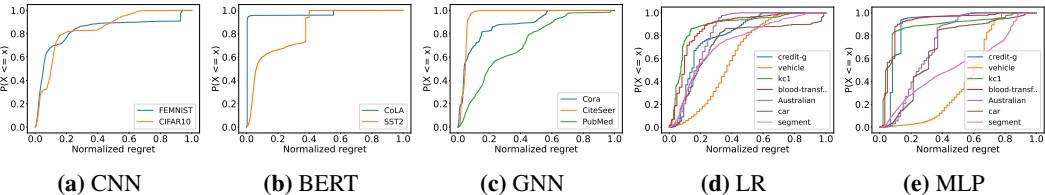

**(a)** CNN  **(b)** BERT  **(c)** GNN  **(d)** LR  **(e)** MLP

**Figure 3:** Empirical Cumulative Distribution Functions: The normalized regret is calculated for all evaluated configurations of the respective model on the respective FL task with FedAvg.

## 3.2 FLEXIBILITY

We allow users to instantiate each benchmark with arguments other than the <dataset, model, algorithm> triple. So, both the underlying objective function and the amount of time it takes to calculate its value can be customized according to the cases specified by the users.

**Objective function.** FEDHPO-BENCH provides the basic interface to support multi-objective optimization concerning arbitrary specified performance metrics and privacy and fairness-related metrics. For privacy protection, we employ a representative FL+DP algorithm NbAFL (Wei et al., 2020) provided by FS, where users can specify any valid value for the privacy budget. As for fairness, FS has provided many personalized FL algorithms and fairness-aware aggregations, and FEDHPO-BENCH can record client-wise performances. In designing the interface of FEDHPO-BENCH, we allow users to specify their preferred measurements of privacy leakage risk and fairness. Then the execution of an FL algorithm can be regarded as evaluating a vector-valued function rather than a scalar-valued one. By default, FEDHPO-BENCH transforms the vector result into a scalar one by treating privacy and fairness-related values as soft constraints to penalize.

**System model.** In addition to customizing the objective function, it is also very helpful to customize the execution time of function evaluation because the execution time of the same FL course can vary a lot when deployed in environments with different system conditions. Many existing HPO benchmarks record the execution time of the training courses, which cannot be adapted regarding users' system conditions. Despite a recorded execution time, we provide a system model to estimate the time consumed by evaluating $f(\lambda, b)$ in realistic scenarios, which is configurable so that users with different system conditions can calibrate the model to their cases (Mohr et al., 2021). Based on the analysis of such a system model and a basic instance (Wang et al., 2021a), we propose and implement our system model, where the execution time $T(f, \lambda, b)$ for each round in evaluating $f(\lambda, b)$ is the summation of time consumed by computation and communication. Roughly, the time for communication is the summation of the time for downloading and uploading transferred information and the latency for establishing connections. The time for computation is the summation of the time for the server's aggregation step and that for the straggler client's local updates. Our system model exposes several adjustable parameters, for which we provide default choices based on the records collected from creating the tabular benchmarks. Meanwhile, users are allowed to specify these parameters according to their scenarios or other system statistic providers, e.g., estimating the latency of stragglers by sampling from FedScale (Lai et al., 2022). We defer the details about our system model to Appendix E.

### 3.3 EXTENSIBILITY

As FedHPO is springing up, we must reduce the effort of introducing more FedHPO problems and novel FedHPO methods to FEDHPO-BENCH.

Recall that a FedHPO problem is characterized by the <dataset, model, algorithm> triple. With FS, we can apply the off-the-shelf data splitters to transform an arbitrary centralized dataset into an FL dataset, reuse any open-sourced model implementation by registering it in FS, and develop a novel FL algorithm via plugging in the hook function that expresses its unique step(s).

Traditional HPO methods are decoupled from the procedure of function evaluation, with a conceptually standard interface for interaction (see Figure 1 and Figure 2). Thus, any new method in this line can readily interplay with FEDHPO-BENCH. However, FedHPO methods, such as FTS and FedEx, are fused with the FL training course to make concurrent exploration, as the dashed blue box in Figure 1 and the red color "FedHPO" module in Figure 2 shows. Thus, we need to implement such methods in FS if we want to benchmark them on FEDHPO-BENCH.

At a high level, such FedHPO methods essentially aim to learn a policy $\pi$ collaboratively, along with the original FL course. As FS is featured by its event-driven programming paradigm, a standard FL course is modularized into event-handler pairs that express all the subroutines. Benefiting from this event-driven paradigm, all we need to develop are augmenting the messages exchanged by FL participants (i.e., re-defining events) and plugging those policy learning-related operations into the event handlers. As a result, we have implemented FTS, FedEx, and a personalized FedEx in FS, where their differences mainly lie in just the definition and implementation of those plug-in operations. We defer more implementation details to Appendix F.

## 4 EXPERIMENTS

We conduct extensive empirical studies with our proposed FEDHPO-BENCH. Basically, we exemplify the use of FEDHPO-BENCH in comparing HPO methods, which, in the meantime, can somewhat validate the correctness of FEDHPO-BENCH. Moreover, we aim to gain more insights into FedHPO, answering two research questions: **(RQ1)** How do traditional HPO methods behave in solving FedHPO problems? **(RQ2)** Do recently proposed FedHPO methods that exploit "concurrent exploration" (see Section 2.2) significantly improve traditional methods? We conduct empirical studies in Section 4.1 and Section 4.2 to answer RQ1 and RQ2, respectively. All scripts concerning the studies here have been committed to FEDHPO-BENCH so that the community can quickly reproduce our established benchmarks.

### 4.1 STUDIES ABOUT APPLYING TRADITIONAL HPO METHODS IN THE FL SETTING

To answer RQ1, we largely follow the experiment conducted in HPOBench (Eggensperger et al., 2021) but focus on the FedHPO problems FEDHPO-BENCH provided.

**Protocol.** We employ up to ten optimizers (i.e., HPO methods) from widely adopted libraries (see Table 6 for more details). For black-box optimizers (*BBO*), we consider random search (*RS*), the evolutionary search approach of differential evolution (*DE* (Storn & Price, 1997; Awad et al., 2020)), and bayesian optimization with: a GP model ($BO_{GP}$), a random forest model ($BO_{RF}$ (Hutter et al., 2011b)), and a kernel density estimator ($BO_{KDE}$ (Falkner et al., 2018b)), respectively. For multi-fidelity optimizers (*MF*), we consider Hyperband (*HB* (Li et al., 2017)), its model-based extensions with KDE-based model (*BOHB* (Falkner et al., 2018a)), and differential evolution (*DEHB* (Awad et al., 2021)), and Optuna's implementations of TPE with median stopping ($TPE_{MD}$) and TPE with Hyperband ($TPE_{HB}$) (Akiba et al., 2019). We apply these optimizers to solve the cross-silo FedHPO problems summarized in Table 1, where the time budget is relaxed for these traditional HPO methods to satisfy multiple full-fidelity function evaluations rather than a one-shot setting. For the sake of efficiency, we conduct this experiment with the *tabular* mode of FEDHPO-BENCH, and consider *#round* as the fidelity dimension for HPO methods to control (while keeping *client_sample_rate*=1.0). To compare the optimizers uniformly and fairly, we repeat each setting five times in the same runtime environment but with different random seeds. The best-seen validation loss is monitored for each optimizer (for multi-fidelity optimizers, higher fidelity results are preferred over lower ones). We sort the optimizers by their best-seen results and compare their mean ranks on all the considered

FedHPO problems. Following HPOBench, we use sign tests to judge (1) whether advanced methods outperform their baselines and (2) whether multi-fidelity methods outperform their single-fidelity counterparts. We refer our readers to Appendix C for more details.

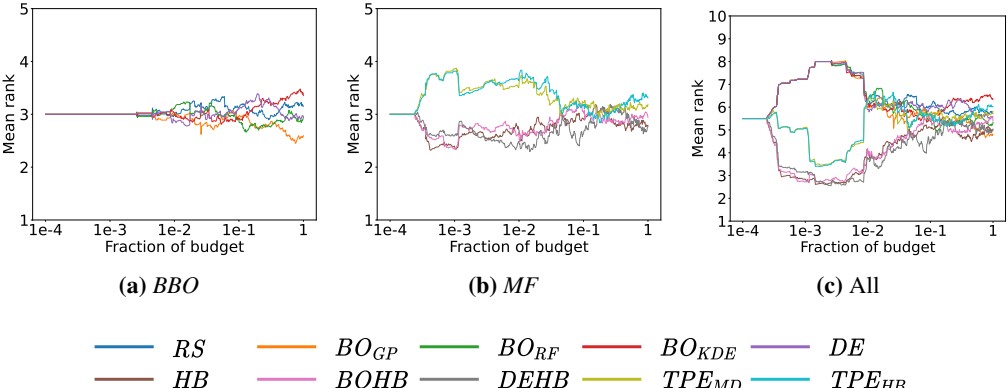

**(a)** *BBO*        **(b)** *MF*        **(c)** All

**Figure 4:** Mean rank over time on all FedHPO problems (with FedAvg).

**Results and Analysis.** We show the overall results in Figure 4, and we defer detailed results to Appendix I. Overall, their eventual mean ranks do not deviate remarkably. For *BBO*, the performances of optimizers are close at the beginning but become more distinguishable along with their exploration. Ultimately, $BO_{GP}$ has successfully sought better configurations than other optimizers. In contrast to *BBO*, *MF* optimizers perform pretty differently in the early stage, which might be rooted in the vast variance of low-fidelity function evaluations. Eventually, *HB* and *BOHB* become superior to others while achieving a very close mean rank. We consider optimizers' final performances on all the considered FedHPO problems, where, for each pair of optimizers, one may win, tie, or lose against the other. Then we can conduct sign tests to compare pairs of optimizers, where results are presented in Table 2 and Table 3. (1) Comparing these advanced optimizers with their baselines, only $BO_{GP}$, $BO_{RF}$, and *DE* win on more than half of the problems but have no significant improvement, which is inconsistent with the non-FL setting. It is worth noting that a similar phenomenon can also be observed for HPO problems in general (Pushak & Hoos, 2022). (2) Meanwhile, no *MF* optimizers show any advantage in exploiting experience, which differs from non-FL cases. We presume the reason lies in the distribution of configurations' performances (see Figure 3). From Table 3, we draw part of the answer to RQ1 as that *MF* optimizers always outperform their corresponding single-fidelity version, which is consistent with the non-FL settings.

**Table 2:** P-value of a sign test for the hypothesis—these advanced methods surpass the baselines.

|  | $BO_{GP}$ | $BO_{RF}$ | $BO_{KDE}$ | DE |
|---|---|---|---|---|
| p-value agains *RS* | 0.0637 | 0.2161 | 0.1649 | 0.7561 |
| win-tie-loss | 13 / 0 / 7 | 12 / 0/ 8 | 7 / 0 / 13 | 11 / 0 / 9 |
|  | BOHB | DEHB | $TPE_{MD}$ | $TPE_{HB}$ |
| p-value against *HB* | 0.4523 | 0.9854 | 0.2942 | 0.2454 |
| win-tie-loss | 7 / 0 / 13 | 9 / 0 / 11 | 9 / 0 / 11 | 9 / 0 / 11 |

**Table 3:** P-value of a sign test for the hypothesis—*MF* methods surpass corresponding *BBO* methods.

|  | *HB* vs. *RS* | *DEHB* vs. *DE* | *BOHB* vs. $BO_{KDE}$ |
|---|---|---|---|
| p-value | 0.1139 | 0.2942 | 0.0106 |
| win-tie-loss | 13 / 0 / 7 | 13 / 0 / 7 | 16 / 0 / 4 |

## 4.2 STUDIES ABOUT CONCURRENT EXPLORATION

To answer RQ2, we select the superior optimizers from Section 4.1 to compare with FedEx (Khodak et al., 2021). As mentioned in Section 2.2, FL allows HPO methods to take advantage of concurrent

exploration, which somewhat compensates for the number of function evaluations. We are interested in methods designed regarding these characteristics of FedHPO and design this experiment to see how much concurrent exploration contributes.

**Protocol.** We consider the FedHPO problem <FEMNIST, CNN, FedAvg>, i.e., FedAvg is applied to learn a 2-layer CNN on FEMNIST. As a full-fidelity function evaluation consumes 500 rounds on this dataset, we specify $RS$, $BO_{GP}$, $BO_{RF}$, $BO_{KDE}$, $HB$, and $BOHB$ to limit their total budget to 2,500 (i.e., 5 times budget of a full-fidelity evaluation) in terms of *#round*. Precisely, each BBO method consists of 50 trials, each of which runs for 50 rounds. For MF optimizers, we set the $\eta$ of Successive Halving Algorithm (SHA) (Jamieson & Talwalkar, 2016) to 3, the minimal budget to 9 rounds, and the max budget to 81 rounds. Then we adopt these optimizers and FedEx wrapped by them (*X+FedEx*) to optimize the design choices of FedAvg, respectively. The wrapper is responsible for determining the arms for each execution of FedEx. We consider validation loss the metric of interest, and function evaluations are conducted in the *raw* mode. We repeat each method three times and report the averaged best-seen value at the end of each trial. Meanwhile, for each considered method, we entirely run the FL course with the optimal configuration it seeks. Their averaged test accuracies are compared.

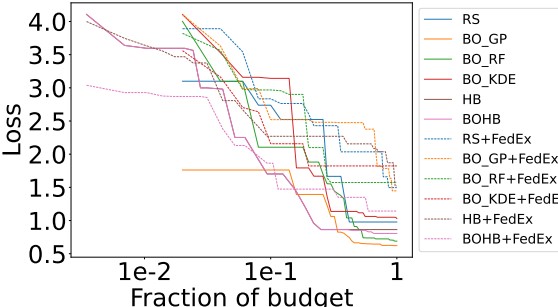

**Figure 5:** Mean validation cross-entropy loss over time.

| Methods | W/O FedEx | W/ FedEx |
|---|---|---|
| *RS* | 79.93 ± 2.45 | 82.03 ± 2.08 |
| *BO_{GP}* | 82.18 ± 0.94 | 83.20 ± 1.24 |
| *BO_{RF}* | 81.86 ± 1.10 | 82.20 ± 0.54 |
| *BO_{KDE}* | 81.34 ± 1.75 | 82.11 ± 0.46 |
| *HB* | 80.26 ± 2.02 | 82.47 ± 0.04 |
| *BOHB* | 79.59 ± 2.09 | **84.02 ± 0.50** |

**Table 5:** Compare the searched configurations: Mean test accuracy (%) ± standard deviation. Underline indicates improvements.

**Results and Analysis.** We present the results in Figure 5 and Table 5. For FedAvg, the best-seen mean validation losses of all wrapped FedEx decrease slower than their corresponding wrapper. However, their searched configurations' generalization performances are significantly better than their wrappers, which strongly confirms the effectiveness of concurrent exploration. Thus, we have a clear answer to RQ2: concurrent exploration methods significantly improve traditional methods.

## 5 CONCLUSION AND FUTURE WORK

In this paper, we first identify the uniqueness of FedHPO, which we ascribe to the distributed nature of FL and its heterogeneous clients. This uniqueness prevents FedHPO research from leveraging existing HPO benchmarks, which has led to inconsistent comparisons between some recently proposed methods. Hence, we suggest and implement a comprehensive, flexible, and extensible benchmark suite, FEDHPO-BENCH. We further conduct extensive HPO experiments on FEDHPO-BENCH, validating its correctness and applicability to comparing traditional and FedHPO methods. We have open-sourced FEDHPO-BENCH with an Apache-2.0 license and will actively maintain it in the future (Maintenance of FEDHPO-BENCH is discussed in Appendix A). We believe FEDHPO-BENCH can serve as the stepping stone to developing reproducible FedHPO works.

In our next step, tasks other than federated supervised learning will be incorporated. At the same time, we aim to extend FEDHPO-BENCH to include different FL settings, e.g., HPO for vertical FL (Zhou et al., 2021). Another issue the current version has not touched on is the risk of privacy leakage caused by HPO methods (Koskela & Honkela, 2020), which we should provide related metrics and testbeds in the future.

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

## A    MAINTENANCE OF FEDHPO-BENCH

In this section, we present our plan for maintaining FEDHPO-BENCH following Eggensperger et al. (2021).

- **Who is maintaining the benchmarking library?** FEDHPO-BENCH is developed and maintained by FedHPO-Bench team (to be de-anonymized later).

- **How can the maintainer of the dataset be contacted (e.g., email address)?** Users can reach out to the maintainer via creating issues on the Github repository with FEDHPO-BENCH label.

- **Is there an erratum?** No.

- **Will the benchmarking library be updated?** Yes, as we discussed in Section 5, we will add more FedHPO problems and introduce more FL tasks to the existing benchmark. We will track updates and Github release on the README. In addition, we will fix potential issues regularly.

- **Will older versions of the benchmarking library continue to be supported/hosted/maintained?** All older versions are available and maintained by the Github release, but limited support will be provided for older versions. Containers will be versioned and available via AliyunOSS.

- **If others want to extend/augment/build on/contribute to the dataset, is there a mechanism for them to do so?** Any contribution is welcome, and all commits to FEDHPO-BENCH must follow the guidance and regulations at `https://github.com/FedHPO-Bench/FedHPO-Bench-ICLR23/blob/main/README.md`.

## B    RELATED WORK

**Hyperparameter Optimization (HPO).** Generally, HPO is an optimization problem where the objective function is non-analytic, non-convex, and even non-differentiable. Therefore, most HPO methods solve such an optimization problem in a trial-and-error manner, with different strategies for balancing exploitation and exploration. Model-free methods such as random search (RS) (Bergstra & Bengio, 2012) and grid search query a set of initially determined hyperparameter configurations without any exploitation. Model-based methods such as Bayesian Optimization (BO) (Shahriari et al., 2016) employ a surrogate model to approximate the objective function. Methods in this line (Hutter et al., 2011a; Lindauer et al., 2022; Hutter et al., 2011b; Falkner et al., 2018b) mainly differ from each other in their surrogate model and how they determine the next query. There are also Evolutionary Algorithms (EAs) that iteratively maintain a population. We consider differential EAs (Storn & Price, 1997; Awad et al., 2020) in our experiments.

As training a deep neural network on a large-scale dataset is very costly, the full-fidelity function evaluations made by BO methods are often unaffordable in practice. Naturally, researchers consider trading off the precision of a function evaluation for its efficiency by, e.g., training fewer epochs and training on a subset of the data. Hyperband (Li et al., 2017) is a representative multi-fidelity method that calls the Successive Halving Algorithm (SHA) (Jamieson & Talwalkar, 2016) again and again with a different number of initial candidates. However, in each execution of SHA, the initial candidates are randomly sampled without any exploitation. To exploit the experience of previous SHA executions, researchers combine BO methods with Hyperband (Falkner et al., 2018a; Awad et al., 2021).

**Benchmarking HPO.** AutoML-related optimization benchmarks have been proved helpful for promoting fair comparisons of related methods and reproducible research works. There have been many successful examples (Hutter et al., 2014; Hansen et al., 2021; Hase et al., 2021; Turner & Eriksson, 2019; Dong & Yang, 2020; Dong et al., 2021; Gijsbers et al., 2019). Noticeably, HPO-B (Pineda-Arango et al., 2021) is highlighted by its support for benchmarking transfer-HPO methods, and HPOBench (Eggensperger et al., 2021) fills the gap of missing multi-fidelity HPO benchmarks.

However, existing HPO benchmarks mainly focus on centralized ML, yet FL, as a promising learning paradigm, has been ignored. In this paper, we identify the uniqueness of FedHPO (see Section 2.2) and implement FEDHPO-BENCH to satisfy the demand for a FedHPO benchmark suite.

**Federated Learning (FL).** In this paper, we restrict our discussion of FL to the "standard" scenario introduced in Section 2.2, where FedAvg (McMahan et al., 2017) is widely adopted. Fancy FL optimization algorithms, including FedProx (Li et al., 2020b) and FedOpt (Asad et al., 2020), are mainly designed to improve the convergence rate and/or better handle the non-IIDness among clients (Wang et al., 2021a). Despite these synchronous optimization algorithms, asynchronous ones (Huba et al., 2022; Xie et al., 2022) are proposed to keep a high concurrency utility.

Sometimes, learning one global model is insufficient to handle the non-IIDness, which calls for personalized FL (Kairouz et al., 2019; Wang et al., 2021a). Many popular pFL algorithms, such as FedBN (Li et al., 2021b) and Ditto (Li et al., 2021a), have been incorporated into FS (Xie et al., 2022), with their unique hyperparameters exposed. Thus, we can further extend FEDHPO-BENCH by considering FedHPO tasks of optimizing the hyperparameters of such algorithms.

**FedHPO.** When we consider HPO in the FL setting, as mentioned in Section 2.2, there is some uniqueness that brings in challenges while, at the same time, it can be leveraged by deliberately designed FedHPO methods. For example, in contrast to traditional HPO methods that query one configuration in each trial, FedEx (Khodak et al., 2021) maintains one policy for determining the client-side hyperparameters and independently samples each client's configuration in each communication round. Different configurations may be evaluated with the same model parameters, which is in analogy to the weight-sharing idea in neural architecture search (NAS) (Liu et al., 2019), as summarized by the authors of FedEx. However, due to the non-IIDness among clients, clients' HPO objective functions tend to be different, where determining their configurations by only one policy might be unsatisfactory. Regarding this issue, FTS (Dai et al., 2020) can be treated as a personalized FedHPO method, where each client maintains its own policy. During the learning procedure, the clients benefit each other by sharing the policies in a privacy-preserving manner and conducting Thompson sampling.

It is worth mentioning that parallel algorithms have been utilized in HPO (Jones, 2001; Hutter et al., 2012). However, in FedHPO, the clients actually do not correspond to the same black-box function due to the heterogeneity among them. Essentially, FedHPO methods instantiate the concurrent exploration idea with extra assumptions. Besides, vanilla parallel HPO methods may leak privacy in the aggregation step, which has been carefully taken into account by FTS.

Dynamic algorithm configuration methods (Biedenkapp et al., 2020; Adriaensen et al., 2022) employ reinforcement learning to learn policies for online adjustments of algorithm parameters, since different parameter values can be optimal at different stages. In contrast to DAC methods, the policy $\pi$ learned in FedEx is responsible for determining the optimal lower-level response of the bi-level optimization problem discussed in Section 2.2, which can be regarded as a multi-armed bandit problem rather than a Markov decision process. In other words, combined with the concurrent exploration strategy, FedEx tries out one arm at a client in each round, where the underlying reward function is assumed to be unchanged across the clients and the whole training course.

As an emerging research topic, existing works relating to FedHPO include Fed-Tuning (Zhang et al., 2021) concerning about system-related performance, learning rate adaptation (Koskela & Honkela, 2020), FLoRA for Gradient Boosted Decision Trees (GBDT), online adaptation scheme-based method (Mostafa, 2020), Auto-FedRL (Guo et al., 2022) for RL-based hyperparameter adaptation, and an insightful comparison between local and global HPO (Holly et al., 2021). These methods can also be easily incorporated into FS, enabling FEDHPO-BENCH to benchmark them.

## B.1 RELATION TO HPOBENCH

HPOBench (Eggensperger et al., 2021) is a collection of multi-fidelity HPO benchmarks, highlighted by their efficiency, reproducibility, and flexibility. These benchmarks can be accessed in either tabular, surrogate, or raw mode. On the one hand, the tabular and surrogate modes enable function evaluation without truly executing the corresponding ML algorithm and thus are efficient. On the other hand, the raw mode means execution in a docker container, which ensures reproducibility. HPOBench provides twelve families of benchmarks that correspond to different data domains, model types, fidelity spaces, etc., and thus flexible usages to validate HPO methods. This collection of HPO benchmarks can promote fair comparisons of related works and reproducible research work, so HPOBench has gained more and more attention from the community.

As pointed out in Section 2.2 that evaluating the objective function that corresponds to an FL algorithm is extremely expensive, FEDHPO-BENCH also prepares tabular and surrogate modes for users to avoid truly executing FL courses. Meanwhile, we provide the raw mode to truly execute an FL course in the docker container of FederatedScope (FS) (Xie et al., 2022).

Sharing the same modes, a question naturally arises—**is it possible to reuse HPOBench's interface for FEDHPO-BENCH?** We answer this question by discussing their commonality and the unique ingredients of FEDHPO-BENCH:

**Commonality.** As the code snippet in Figure 2 shows, the instantiation of a benchmark class, the "ConfigSpace" package based specification of search space, and the protocol for the interaction between an HPO method and a benchmark object are roughly consistent with HPOBench.

**Uniqueness.** In addition to a collection of benchmarks, FEDHPO-BENCH is flexible in terms of enabling users to tailor one benchmark to their scenarios (see Section 3.2). To this end, users are allowed to instantiate a specific benchmark object with extra optional arguments:

- *Privacy budget* with which function evaluation corresponds to the execution of NbAFL (Wei et al., 2020) instead of vanilla FedAvg. Taking the tabular mode, for example, this means looking up a privacy budget-specific table.

- *The type of fairness metric and its strength* with which FEDHPO-BENCH will consider a vector-valued objective function (i.e., client-wise results) rather than a scalar-valued objective function. Besides, the return value of calling the function evaluation will be the mean performance regularized by the specified fairness regularizer.

- *The parameter(s) for our system model* with which the execution time is estimated regarding the user's system condition. Without using a system model, FEDHPO-BENCH can provide the recorded execution time in the creation of this benchmark.

Currently, we implement the interfaces of FEDHPO-BENCH by ourselves, where the style of our interfaces is kept similar to HPOBench for the convenience of users who are familiar with HPOBench. We also provide several examples (`https://github.com/FedHPO-Bench/FedHPO-Bench-ICLR23/tree/main/demo`) to access our tabular, surrogate, and raw benchmarks by implementing HPOBench's abstract base class. As a first step, we are going to contribute more such subclasses to the repository of HPOBench so that users can access our benchmarks via HPOBench's interfaces, where flexible customization cannot be provided temporarily. In our next step, we plan to extend the interfaces of HPOBench such that the benchmarks of FEDHPO-BENCH can be accessed with our proposed flexible customization.

## C   HPO METHODS

As shown in Table 6, we provide an overview of the optimizers (i.e., HPO methods) we use in this paper.

**Table 6:** Overview of the optimizers from widely adopted libraries.

| Name | Model | Packages | version |
|------|-------|----------|---------|
| *RS* (Bergstra & Bengio, 2012) | - | *HPBandster* | 0.7.4 |
| *BO$_{GP}$* (Hutter et al., 2011a; Lindauer et al., 2022) | *GP* | *SMAC3* | 1.3.3 |
| *BO$_{RF}$* (Hutter et al., 2011b) | *RF* | *SMAC3* | 1.3.3 |
| *BO$_{KDE}$* (Falkner et al., 2018b) | *KDE* | *HPBandster* | 0.7.4 |
| *DE* (Storn & Price, 1997; Awad et al., 2020) | - | *DEHB* | git commit |
| *HB* (Li et al., 2017) | - | *HPBandster* | 0.7.4 |
| *BOHB* (Falkner et al., 2018a) | *KDE* | *HPBandster* | 0.7.4 |
| *DEHB* (Awad et al., 2021) | - | *DEHB* | git commit |
| *TPE$_{MD}$* (Akiba et al., 2019) | *TPE* | *Optuna* | 2.10.0 |
| *TPE$_{HB}$* (Akiba et al., 2019) | *TPE* | *Optuna* | 2.10.0 |

## C.1 BLACK-BOX OPTIMIZERS

**RS** (*Random search*) is a priori-free HPO method, i.e., each step of the search does not exploit the already explored configuration. The random search outperforms the grid search within a small fraction of the computation time.

**$BO_{GP}$** is a Bayesian optimization with a Gaussian process model. $BO_{GP}$ uses a Matérn kernel for continuous hyperparameters, and a hamming kernel for categorical hyperparameters. In addition, the acquisition function is expected improvement (EI).

**$BO_{RF}$** is a Bayesian optimization with a random forest model. We set the hyperparameters of the random forest as follows: the number of trees is 10, the max depth of each tree is 20, and we use the default setting of the minimal samples split, which is 3.

**$BO_{KDE}$** is a Bayesian optimization with kernel density estimators (KDE), which is used in *BOHB* (Falkner et al., 2018a). It models objective function as $\Pr(x \mid y_{\text{good}})$ and $\Pr(x \mid y_{\text{bad}})$. We set the hyperparameters for $BO_{KDE}$ as follows: the number of samples to optimize EI is 64, and $1/3$ of purely random configurations are sampled from the prior without the model; the bandwidth factor is 3 to encourage diversity, and the minimum bandwidth is 1e-3 to keep diversity.

**DE** uses the evolutionary search approach of Differential Evolution. We set the mutation strategy to *rand1* and the binomial crossover strategy to *bin* [3]. In addition, we use the default settings for the other hyperparameters of *DE*, where the mutation factor is 0.5, crossover probability is 0.5, and the population size is 20.

## C.2 MULTI-FIDELITY OPTIMIZERS

**HB** (*Hyperband*) is an extension on top of successive halving algorithms for the pure-exploration nonstochastic infinite-armed bandit problem. Hyperband makes a trade-off between the number of hyperparameter configurations and the budget allocated to each hyperparameter configuration. We set $\eta$ to 3, which means only a fraction of $1/\eta$ of hyperparameter configurations goes to the next round.

**BOHB** combines *HB* with the guidance and guarantees of convergence of Bayesian optimization with kernel density estimators. We set the hyperparameter of the *BO* components and the *HB* components of *BOHB* to be the same as $BO_{KDE}$ and *HB* described above, respectively.

**DEHB** combines the advantages of the bandit-based method *HB* and the evolutionary search approach of *DE*. The hyperparameter of *DE* components and *BO* components are set to be exactly the same as *DE* and *HB* described above, respectively.

**$TPE_{MD}$** is implemented in *Optuna* and uses Tree-structured Parzen Estimator (*TPE*) as a sampling algorithm, where on each trial, TPE fits two Gaussian Mixture models for each hyperparameter. One is to the set of hyperparameters with the best performance, and the other is to the remaining hyperparameters. In addition, it uses the median stopping rule as a pruner, which means that it will prune if the trial's best intermediate result is worse than the median (*MD*) of intermediate results of previous trials at the same step. We use the default settings for both *TPE* and *MD*.

**$TPE_{HB}$** is similar to $TPE_{MD}$ described above, which uses *TPE* as a sampling algorithm and *HB* as pruner. We set the reduction factor to 3 for *HB* pruner, and all other settings use the default ones.

## D DATASETS

As shown in Table 7, we provide a detailed description of the datasets we use in current FEDHPO-BENCH. For comprehensiveness, we use 16 FL datasets from 5 domains, including CV, NLP, graph, tabular, and recommendation (Xie et al., 2022; Wang et al., 2022; Eggensperger et al., 2021). Some of them are inherently real-world FL datasets, while others are simulated FL datasets split by the splitter modules of FS. Notably, the name of datasets from OpenML is the ID of the corresponding task.

---

[3]Please refer to `https://github.com/automl/DEHB/blob/master/README.md` for details.

**Table 7:** Statistics of the datasets used in current FEDHPO-BENCH.

| Name | #Client | Subsample | #Instance | #Class | Split by |
|------|---------|-----------|-----------|--------|----------|
| FMNIST | 3,550 | 5% | 805,263 | 62 | Writer |
| CIFAR-10 | 5 | 100% | 60,000 | 10 | LDA |
| CoLA | 5 | 100% | 10,657 | 2 | LDA |
| SST-2 | 5 | 100% | 70,042 | 2 | LDA |
| Cora | 5 | 100% | 2,708 | 7 | Community |
| CiteSeer | 5 | 100% | 4,230 | 6 | Community |
| PubMed | 5 | 100% | 19,717 | 3 | Community |
| Hetero-task | 5 | 100% | 6,760 | 2~6 | Task |
| credit-g$_{31}$ | 5 | 100% | 1,000 | 2 | LDA |
| vehicle$_{53}$ | 5 | 100% | 846 | 4 | LDA |
| kc1$_{3917}$ | 5 | 100% | 2,109 | 2 | LDA |
| blood-transf..$_{10101}$ | 5 | 100% | 748 | 2 | LDA |
| Australian$_{146818}$ | 5 | 100% | 690 | 2 | LDA |
| car$_{146821}$ | 5 | 100% | 1,728 | 4 | LDA |
| segment$_{146822}$ | 5 | 100% | 2,310 | 7 | LDA |
| FedNetflix | 480,189 | 100% | ≈100,000,000 | 5 | User |
| Twitter | 660,120 | 0.5% | 1,600,498 | 2 | User |

**FEMNIST** is an FL image dataset from LEAF (Caldas et al., 2018), whose task is image classification. Following (Caldas et al., 2018), we use a subsample of FEMNIST with 200 clients, which is round 5%. And we use the default train/valid/test splits for each client, where the ratio is 60% : 20% : 20%.

**CIFAR-10** (Krizhevsky et al., 2009) is from Tiny Images dataset and consists of 60,000 $32 \times 32$ color images, whose task is image classification. We split images into 5 clients by latent dirichlet allocation (LDA) to produce statistical heterogeneity among these clients. We split the raw training set to training and validation sets with a ratio $4 : 1$, so that ratio of final train/valid/test splits is 66.7%:16.67%:16.67%.

**SST-2** is a dataset from GLUE (Wang et al., 2018) benchmark, whose task is binary sentiment classification for sentences. We also split these sentences into 5 clients by LDA. In addition, we use the official train/valid/test splits for SST-2.

**CoLA** is also a dataset from GLUE benchmark, whose task is binary classification for sentences—whether it is a grammatical English sentence. We exactly follow the experimental setup in SST-2.

**Cora & CiteSeer & PubMed** (Sen et al., 2008; Yang et al., 2016) are three widely adopted graph datasets, whose tasks are node classification. Following FS-G (Wang et al., 2022), a community splitter is applied to each graph to generate five subgraphs for each client. We also split the nodes into train/valid/test sets, where the ratio is 60%:20%:20%.

**Hetero-task** is a graph classification dataset adopted from Graph-DC (Yao et al., 2022), which contains 5 clients. Each client has different but similar graph classification task, such as molecular attribute prediction. In addition, we set the ratio of train/valid/test splits in each client to 80%:10%:10%.

**Tabular datasets** are consist of 7 tabular datasets from OpenML (Bischl et al., 2017), whose task ids (name of source data) are 31 (**credit-g**), 53 (**vehicle**), 3917 (**kc1**), 10101 (**blood-transfusion-service-center**), 146818 (**Australian**), 146821 (**car**) and 146822 (**segment**). We split each dataset into 5 clients by LDA, respectively. In addition, we set the ratio of train/valid/test splits to 80%:10%:10%.

**FedNetflix** is a recommendation dataset from The Netflix Prize (Bennett & Lanning, 2007), whose task is to predict the ratings between users and movies. Netflix consists of around 100 million ratings between 480,189 users and 171,770 movies. We split the Netflix dataset into 480,189 clients by users. In addition, we set the ratio of train/valid/test splits to 80%:10%:10%.

**Twitter** is a sentiment analysis dataset from LEAF (Caldas et al., 2018), whose task is to determine sentiment of sentences. We use a subsample of Twitter with around 3300 clients. Moreover, we use the train/valid/test splits for each client, where the ratio is 80% : 10% : 10%. It is worth noting that

the average number of samples is only 1.94, which means some clients do not have valid split or test split, and we evaluate the performance on a shared test split merged by all clients.

## E  System Model

In this section, we will discuss the system model in detail we have proposed and implemented. The total execution time of FL consists of the time consumed by communication and the time consumed by calculation, thus, the system model is as follows:

$$
\begin{aligned}
T(f, \lambda, b) &= T_{\text{comm}}(f, \lambda, b) + T_{\text{comp}}(f, \lambda, b), \\
T_{\text{comm}}(f, \lambda, b) &= \frac{S_{\text{down}}(f, \lambda)}{B_{\text{down}}} + \frac{S_{\text{up}}(f, \lambda)}{B_{\text{up}}} + \alpha(N), \\
T_{\text{comp}}(f, \lambda, b) &= \mathbb{E}_{T_i^{(\text{client})} \sim \text{Exp}(\cdot | \frac{1}{c(f, \lambda, b)}), i=1, \ldots, N}[\max(\{T_i^{(\text{client})}\})] + T^{(\text{server})}(f, \lambda, b),
\end{aligned}
\tag{1}
$$

where $N$ denotes the number of clients sampled in this round, $\alpha(N)$ denotes the latency, which is an increasing function of $N$ but is independent of the message size (contains the time needed to establish the transmission between the server and the clients), $S(f, \lambda)$ denotes the download/upload size, $B$ denotes the download/upload bandwidth of client, $T^{(\text{server})}$ is the time consumed by server-side computation, and $T_i^{(\text{client})}$ denotes the computation time consumed by $i$-th client, which is sampled from an exponential distribution with $c(f, \lambda, b)$ as its mean. This design intends to simulate the heterogeneity among clients' computational capacity, where the assumed exponential distribution has been widely adopted in system designs (Wang et al., 2021a) and is consistent with real-world applications (Huba et al., 2022).

We provide default parameters of our system model, including $c(f, \lambda, b)$, $B_{\text{up}}$, $B_{\text{down}}$, and $T^{(\text{server})}$, based on observations collected from FL trials we have conducted and real-world network bandwidth. Users are allowed to specify these parameters according to their scenarios or other system statistic providers, e.g., estimating the computation time of stragglers by sampling from FedScale (Lai et al., 2022). As for the network bandwidth, we set $B_{\text{down}} \sim 0.75\text{MB/secs}$, $B_{\text{up}} \sim 0.25\text{MB/secs}$ following (Lai et al., 2022; Wang et al., 2021a). The default value of $c(f, \lambda, b)$ is obtained by averaging the recorded client-wise time costs in trials of tabular mode benchmarks. Due to the limit on the number of ports of the server, we set the default value of the maximum number of connections in calculating $\alpha(N)$ to 65535.

To implement our system model, we use the following proposition to calculate Eq. 1 analytically, where we use $c$ as a shorthand for $c(f, \lambda, b)$ to keep clarity.

**Proposition 1.** *When the computation time of clients is identically independently distributed, following an exponential distribution $Exp(\cdot | \frac{1}{c})$, then the expected time for the straggler of $N$ uniformly sampled clients is $\sum_{i=1}^{N} \frac{c}{i}$.*

What we need to calculate is the expected maximum of i.i.d. exponential random variable. Proposition 1 states that, for $N$ exponential variables independently drawn from $\text{Exp}(\cdot | \frac{1}{c})$, the expectation is $\sum_{i=1}^{N} \frac{c}{i}$. There are many ways to prove this useful proposition, and we provide a proof starting from studying the minimum of the exponential random variables.

*Proof.* At first, the minimum of $N$ such random variables obeys $\text{Exp}(\cdot | \frac{N}{c})$ (Graham et al., 1989). Denoting the $i$-th minimum of them by $T_i$, $T_1 \sim \text{Exp}(\cdot | \frac{N}{c})$ and $T_N$ is what we are interested in. Meanwhile, it is well known that exponential distribution is memoryless, namely, $\Pr(X > s + t | X > s) = \Pr(X > t)$. Thus, $T_2 - T_1$ obeys the same distribution as the minimum of $N - 1$ such random variables, that is to say, $T_2 - T_1$ is a random variable drawn from $\text{Exp}(\cdot | \frac{N-1}{c})$. Similarly, $(T_{i+1} - T_i) \sim \text{Exp}(\cdot | \frac{N-i}{c}), i = 1, \ldots, N - 1$. Thus, we have:

$$
\mathbb{E}[T_N] = \mathbb{E}[T_1 + \sum_{i=1}^{N-1}(T_{i+1} - T_i)] = \frac{c}{N} + \sum_{i=1}^{N-1} \frac{c}{N-i} = \sum_{i=1}^{N} \frac{c}{i},
\tag{2}
$$

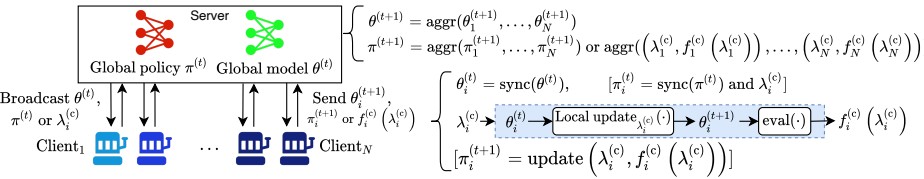

**Figure 6:** A general algorithmic view for FedHPO methods: They are allowed to concurrently explore different client-side configurations in the same round of FL, but the clients are heterogeneous, i.e., corresponding to different functions $f_i^{(c)}(\cdot)$. Operators in brackets are optional.

which concludes this proof. $\qquad\square$

It is worth noting that we provide several optional system models. For example, for point-to-point transport protocols, $T_{\text{comm}}$ should contain the time the server sends the model to each client.

## F    DETAILS OF THE IMPLEMENTATIONS OF FEDEX AND FTS

We first present a general algorithmic view in Figure 6, which unifies several such methods as well as their personalized counterparts. At a high level, a policy $\pi$ for determining the optimal lower-level response $\lambda^{(c)*} = \min_{\lambda^{(c)}} f(\lambda^{(s)}, \lambda^{(c)})$ is to be federally learned, along with the FL course itself. In the $t$-th communication round: (1) In addition to the model $\theta^{(t)}$, either the policy $\pi^{(t)}$ or its decisions $\lambda_i^{(c)}$ is also broadcasted. (2) For the $i$-th client, if $\pi^{(t)}$ is received, it needs to synchronize its local policy $\pi_i^{(t)}$ with this global one and then sample a hyperparameter configuration $\lambda_i^{(c)}$ from its local policy. (3) Either received or locally sampled, $\lambda_i^{(c)}$ is used to specify the local update procedure of FL, which results in updated local model $\theta_i^{(t+1)}$. (4) Then $\theta_i^{(t+1)}$ is evaluated to provide the result of (client-specific) function evaluation $f_i^{(c)}\left(\lambda_i^{(c)}\right)$. (5) For personalized FedHPO methods that maintain a local policy $\pi_i$, it is updated w.r.t. $(\lambda_i, f_i(\lambda_i))$ to produce $\pi_i^{(t+1)}$. (6) In addition to the local model $\theta_i^{(t+1)}$, either the local policy $\pi_i^{(t+1)}$ or the feedback $\left(\lambda_i^{(c)}, f_i^{(c)}\left(\lambda_i^{(c)}\right)\right)$ is sent to the server. (7) Finally, the server aggregates $\theta_i^{(t+1)}$s into $\theta^{(t+1)}$ and $\pi_i^{(t+1)}$s/$\left(\lambda_i^{(c)}, f_i^{(c)}\left(\lambda_i^{(c)}\right)\right)$ s into $\pi^{(t+1)}$, respectively.

In FedEx (Khodak et al., 2021), $\lambda_i$s are independently sampled from $\pi$, and the aggregation operator "aggr$_\text{p}$" is exponential gradient descent. In FTS (Dai et al., 2020), the broadcasted policy $\pi^{(t)}$ is the samples drawn from all clients' posterior beliefs. The synchronous operator "sync$_\text{p}$" can be regarded as mixing Gaussian process (GP) models. The update operator "update$_\text{p}$" corresponds to updating local GP model. Then a sample drawn from local GP posterior belief is regarded as $\pi_i^{(t+1)}$ and uploaded. Finally, the aggregation operator "aggr$_\text{p}$" is packing received samples together.

## G    STUDIES ABOUT THE NEW FIDELITY

In FL, a larger *client_sample_rate* leads to a minor variance of the aggregated model in each round, which is believed to need less *#round* for convergence and to perform better. Therefore, we tend to set the *client_sample_rate* as close to 1 as possible. However, according to our system model in Section 3.2, a large *client_sample_rate* leads to an increase in latency ($\alpha(N)$), which makes the communication cost higher. To answer RQ3, we use tabular mode and study the trade-off between these two fidelity dimensions: *client_sample_rate* and *#round*. We simulate two distinct system conditions by specifying different parameters for our system model.

**Protocol.** We compare the performance of *HB* with different *client_sample_rate*s to learn a 2-layer CNN with 2,048 hidden units on FEMNIST. To simulate a system condition with bad network status, we set the upload bandwidth $B_{\text{up}}$ to 0.25MB/second and the download bandwidth $B_{\text{down}}$ to 0.75MB/second (Wang et al., 2021a). As for good network status, we set the upload bandwidth $B_{\text{up}}$ to 0.25GB/second and the download bandwidth $B_{(\text{down})}$ to 0.75GB/second. In both cases, we consider

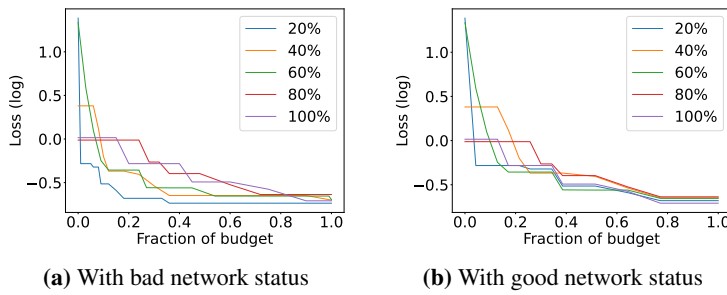

**Figure 7:** Performances of different *client_sample_rate* under different system conditions.

different computation overhead so that it is negligible and significant, respectively. As for the rest settings, we largely follow that in Section 4.1.

**Results and Analysis.** We have an answer to RQ3: with the same time budget, the FL procedure with a lower *client_sample_rate* achieves a better result than higher *client_sample_rate* with the bad network status. In comparison, that with a higher *client_sample_rate* achieves a better result than lower *client_sample_rate* in the good network status. In conclusion, this study suggests a best practice of pursuing a more economic accuracy-efficiency trade-off by balancing *client_sample_rate* with *#round*, w.r.t. the system condition. Better choices tend to achieve more economical accuracy-efficiency trade-offs for FedHPO.

# H    DETAILS ON FEDHPO-BENCH BENCHMARKS

FEDHPO-BENCH consists of serveral categories of benchmarks on the different datasets (see Appendix D) with three modes. If not specified, we use the model as the name of the benchmark in cross-silo scenario. In this part, we provide more details about how we construct the FedHPO problems provided by current FEDHPO-BENCH and the three modes to interact with them.

## H.1    CATEGORY

We categorize our benchmarks by model types. Each benchmark is designed to solve specific FL HPO problems on its data domain, wherein CNN benchmark on CV, BERT benchmark on NLP, GNN benchmark on the graphs, and LR & MLP benchmark on tabular data. All benchmarks have several hyperparameters on configuration space and two on fidelity space, namely sample rate of FL and FL round. And the benchmarks support several FL algorithms, such as FedAvg and FedOpt.

**CNN** benchmark learns a two-layer CNN with 2048 hidden units on FEMNIST and 128 hidden units on CIFAR-10 with five hyperparameters on configuration space that tune the batch size of the dataloader, the weight decay, the learning rate, the dropout of the CNN models, and the step size of local training round in client each FL communication round. The tabular and surrogate mode of the CNN benchmark only supports FedAvg due to our limitations in computing resources for now, but we will update FEDHPO-BENCH with more results as soon as possible.

**BERT** benchmark fine-tunes a pre-trained language model, BERT-Tiny, which has two layers and 128 hidden units, on CoLA and SST-2. The BERT benchmark also has five hyperparameters on configuration space which is the same as CNN benchmark. In addition, the BERT benchmark support FedAvg and FedOpt with all three mode.

**GNN** benchmark learns a two-layer GCN with 64 hidden units on Cora, CiteSeer and PubMed. The GNN benchmark has four on hyperparameters configuration space that tune the weight decay, the learning rate, the dropout of the GNN models, and the step size of local training round in client each FL communication round. The GNN benchmark support FedAvg and FedOpt with all three mode.

**Hetero** benchmark learns a two-layer GCN with 64 hidden units as backbone to be aggregated. The Hetero benchmark has two on hyperparameters configuration space that tune the learning rate and the step size of local training round in client each FL communication round. For each client, there are

**Table 8:** The search space of our benchmarks, where continuous search spaces are discretized into several bins under the tabular mode.

| Benchmark | | Name | Type | Log | #Bins | Range |
|---|---|---|---|---|---|---|
| CNN | Client | batch_size | int | × | - | {16, 32, 64} |
| | | weight_decay | float | × | 4 | [0, 0.001] |
| | | dropout | float | × | 2 | [0, 0.5] |
| | | step_size | int | × | 4 | [1, 4] |
| | | learning_rate | float | ✓ | 10 | [0.01, 1.0] |
| | Server | momentum | float | × | 2 | [0.0, 0.9] |
| | | learning_rate | float | × | 3 | [0.1, 1.0] |
| | Fidelity | client_sample_rate | float | × | 5 | [0.2, 1.0] |
| | | round | int | × | 250 | [1, 500] |
| BERT | Client | batch_size | int | × | - | {8, 16, 32, 64, 128} |
| | | weight_decay | float | × | 4 | [0, 0.001] |
| | | dropout | float | × | 2 | [0, 0.5] |
| | | step_size | int | × | 4 | [1, 4] |
| | | learning_rate | float | ✓ | 10 | [0.01, 1.0] |
| | Server | momentum | float | × | 2 | [0.0, 0.9] |
| | | learning_rate | float | × | 3 | [0.1, 1.0] |
| | Fidelity | client_sample_rate | float | × | 5 | [0.2, 1.0] |
| | | round | int | × | 40 | [1, 40] |
| GNN | Client | weight_decay | float | × | 4 | [0, 0.001] |
| | | dropout | float | × | 2 | [0, 0.5] |
| | | step_size | int | × | 8 | [1, 8] |
| | | learning_rate | float | ✓ | 10 | [0.01, 1.0] |
| | Server | momentum | float | × | 2 | [0.0, 0.9] |
| | | learning_rate | float | × | 3 | [0.1, 1.0] |
| | Fidelity | client_sample_rate | float | × | 5 | [0.2, 1.0] |
| | | round | int | × | 500 | [1, 500] |
| Hetero | Client | learning_rate | float | ✓ | 2 | [0.001, 0.01] |
| | | step_size | int | × | 2 | [1, 4] |
| | Fidelity | client_sample_rate | float | × | 5 | [0.2, 1.0] |
| | | round | int | × | 500 | [1, 500] |
| LR | Client | batch_size | int | ✓ | 7 | [4, 256] |
| | | weight_decay | float | × | 4 | [0, 0.001] |
| | | step_size | int | × | 4 | [1, 4] |
| | | learning_rate | float | ✓ | 6 | [0.00001, 1.0] |
| | Server | momentum | float | × | 2 | [0.0, 0.9] |
| | | learning_rate | float | × | 3 | [0.1, 1.0] |
| | Fidelity | client_sample_rate | float | × | 5 | [0.2, 1.0] |
| | | round | int | × | 500 | [1, 500] |
| MLP | Client | batch_size | int | ✓ | 7 | [4, 256] |
| | | weight_decay | float | × | 4 | [0, 0.001] |
| | | step_size | int | × | 4 | [1, 4] |
| | | learning_rate | float | ✓ | 6 | [0.00001, 1.0] |
| | | depth | int | × | 3 | [1, 3] |
| | | width | int | ✓ | 7 | [16, 1024] |
| | Server | momentum | float | × | 2 | [0.0, 0.9] |
| | | learning_rate | float | × | 3 | [0.1, 1.0] |
| | Fidelity | client_sample_rate | float | × | 5 | [0.2, 1.0] |
| | | round | int | × | 500 | [1, 500] |

personalized encoder and classifier to handle different tasks. Thus, the Hadamard product of each client's configuration makes the search space.

**LR** benchmark learns an lr on seven tasks from OpenML, see Appendix D for details. The LR benchmark has four on hyperparameters configuration space that tune the batch size of the dataloader, the weight decay, the learning rate, and step size of local training round in client each FL communication round. The LR benchmark support FedAvg and FedOpt with all three mode.

**MLP** benchmark's the vast majority of settings are the same as LR benchmark. But in particular, we add depth and width of the MLP to search space in terms of model architecture. The MLP benchmark also support FedAvg and FedOpt with all three mode.

**Cross-device**. In cross-device scenarios, there can be large number of clients in total, but only a few participate in each communication round. This benchmark contains two datasets, Twitter and FedNetflix. We use a bag of words model with LR and tune the *learning_rate*, *weight_decay*, and *step_size* of local training round in Twitter. As for FedNetflix, we tune an HMFNet (Li et al., 2021c) in the *learning_rate*, *batch_size*, and *step_size* of local training round. Due to the time limit, the results FedNetflix is incomplete, and we present the ECDF of Twitter in Figure 9.

### H.2 MODE

Following HPOBench (Eggensperger et al., 2021), FEDHPO-BENCH provides three different modes for function evaluation: the tabular mode, the surrogate mode, and the raw mode. The valid input hyperparameter configurations and the speed of acquiring feedback vary from mode to mode. Users can choose the desired mode according to the purposes of their experiments.

**Tabular mode**. The idea is to evaluate the performance of many different hyperparameter configurations in advance so that users can acquire their results immediately. For efficient function evaluation, we implement the tabular mode of FEDHPO-BENCH by running the FL algorithms configured by the grid search space in advance from our original search space (see Table 8). For hyperparameters whose original search space is discrete, we just preserve its original one. As for continuous ones, we discretize them into several bins (also see Table 8 for details). To ensure that the results are reproducible, we execute the FL procedure in the Docker container environment. Each specific configuration $\lambda$ is repeated three times with different random seeds, and the resulted performances, including loss, accuracy and f1-score under train/validation/test splits, are averaged and adopted as the results of $f(\lambda)$. Users can choose the desired metric as the output of the black-box function via FEDHPO-BENCH's APIs. Besides, we provide not only the results of $f(\lambda)$ (i.e., that with full-fidelity) but also results of $f(\lambda, b)$, where $b$ is enumerated across different *#round* and different *client_sample_rate*. Since executing function evaluation is much more costly in FL than traditional centralized learning, such lookup tables are precious. In creating them, we spent about two months of computation time on six machines, each with four Nvidia V100 GPUs. Now we make them publicly accessible via the tabular mode of FEDHPO-BENCH.

**Surrogate mode**.

As tabular mode has discretized the original search space and thus cannot respond to queries other than the grids, we train random forest models on these lookup tables, i.e., $\{(\lambda, b), f(\lambda, b))\}$. These models serve as a surrogate of the functions to be optimized and can answer any query $\lambda$ by simply making an inference. Specifically, we conduct 10-fold cross-validation to train and evaluate random forest models (implemented in scikit-learn (Pedregosa et al., 2011)) on the tabular data. Meanwhile, we search for suitable hyperparameters for the random forest models with the number of trees in $\{10, 20\}$ and the max depth in $\{10, 15, 20\}$. The mean absolute error (MAE) of the surrogate model w.r.t. the true value is within an acceptable threshold. For example, in predicting the true average loss on the CNN benchmark, the surrogate model has a training error of 0.00609 and a testing error of 0.00777. In addition to the off-the-shelf surrogate models we provide, FEDHPO-BENCH offers tools for users to build brand-new surrogate models. Meanwhile, we notice the recent successes of neural network-based surrogate, e.g., YAHPO Gym (Pfisterer et al., 2022), and we will also try it in the next version of FEDHPO-BENCH.

**Raw mode**. Both of the above modes, although they can respond quickly, are limited to pre-designed search space. Thus, we introduce raw mode to FEDHPO-BENCH, where user-defined search spaces are allowed. Once FEDHPO-BENCH's APIs are called with specific hyperparameters, a containerized

and standalone FL procedure (supported by FS) will be launched. It is worth noting that although we use standalone simulation to eliminate the communication cost, raw mode still consumes much more computation cost than tabular and surrogate modes.

### H.3 NEW HYPERPARAMETERS

The FL setting introduces new hyperparameters such as server-side *learning_rate*, *momentum* for FedOpt (Asad et al., 2020) and client-side *#local_update_step*. Different FL algorithms have different parameters, which correlate with hyperparameters related to general ML procedures. In this section, we first adopt FedProx (Li et al., 2020b) to study the impact of server-side hyperparameters $mu$, the coefficient of the regular term, on the results. And then we compare the landscape of the federated learning method and non-federated method.

#### H.3.1 TRENDS WITH DIFFERENT REGULARITY IN FEDPROX

To extend the tabular benchmarks with more FL algorithms, we adopt FedProx (Li et al., 2020b) to GNN benchmark. Based on Table 8, we tune the server-side hyperparameters $mu$, the coefficient of the regular term, in {0.1, 1.0, 5.0} to study the trends with different regularity in FedProx. We show the landscape in Figure 10 with learning rate in [0.01, 1.0] and $mu$ in [0.1, 5.0], and we observe that when the learning rate is low, the effect of mu has little impact on the accuracy; however, when the learning rate is large, the increase of mu can seriously damage the accuracy.

#### H.3.2 LANDSCAPES ON ML-RELATED HYPERPARAMETERS

In this section, to study the validation loss landscape of the federated learning method (FedAvg) and non-federated method (Isolated), we consider *learning_rate* and *batch_size*, the hyperparameters of the ML algorithm, as the coordinate axis to build the loss landscapes. We fix other ML-related hyperparameters *weight_decay* to 0.0, *dropout* to 0.5 for both FedAvg and Isolated, which is the best configuration chosen from the tabular benchmark <CNN, FEMNIST, FedAvg> under 1.0 *client_sample_rate*. As the loss landscapes shown in Figure 8 with learning rate in [0.01, 1.0] and batch size in 16, 32, 64, we observe that the FedAvg with a higher learning rate achieves better results, while the non-federated method (Isolated) prefer a lower learning rate. Their differences suggest the uniqueness of FedHPO's objective functions.

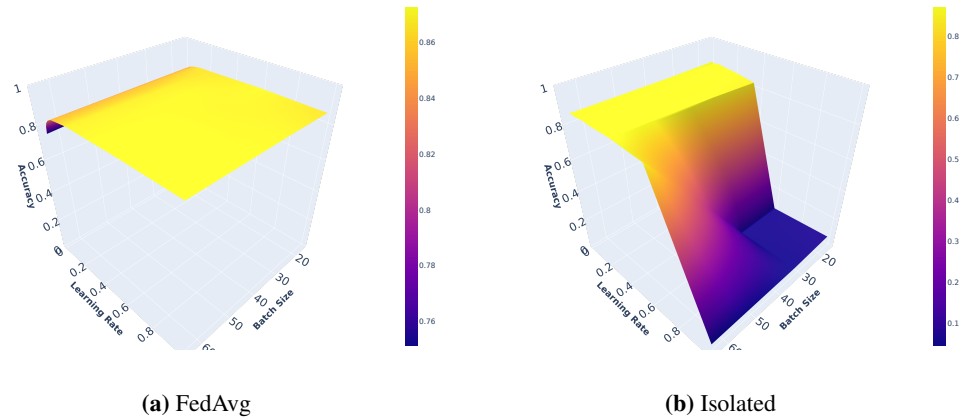

(a) FedAvg                    (b) Isolated

**Figure 8:** Landscape with the hyperparameters of the ML algorithm on FEMNIST.

### H.4 DATA ANALYTICS

#### H.4.1 TRENDS IN DIFFERENT PRIVACY BUDGETS

We extend the tabular benchmarks with different levels of privacy budgets in FEMNIST and Cora. To explore the trends of optimal configurations under different privacy budgets, we adopt NbAFL (Wei

et al., 2020) with $\epsilon \in \{1, 10, 20\}$. We observe that the best configuration varies under different levels of privacy budgets in and Cora, as shown in Table 10. Under different privacy budgets, a large *step_size* all leads to a good performance. However, when the noise is intense, a higher *learning_rate* is preferred, while a lower *learning_rate* will perform better when the noise is weak.

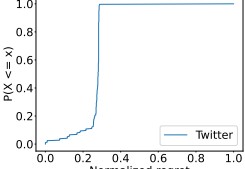

| $\epsilon$ | learning_rate | weight_decay | dropout | step_size | Test Acc. (%) |
|---|---|---|---|---|---|
| 1 | 1.0 | 0.001 | 0.5 | 7 | 63.87 ± 6.38 |
| 10 | 0.59948 | 0.0 | 0.5 | 6 | 87.02 ± 1.16 |
| 20 | 0.59948 | 0.001 | 0.5 | 6 | 87.30 ± 0.54 |

**Figure 9:** ECDF on Twitter.     **Table 10:** Best configuration with different levels of privacy budgets in Cora.

### H.4.2    ERRORS OF SURROGATE BENCHMARKS

As we mentioned in Section H.2, we report the regression error of training surrogate model in Table 12. Meanwhile, we present the mean rank over time of optimizers with surrogate modes in Figure 22 and Figure 23. Compared to the results of tabular modes in 13 and 14, $BO_{GP}$ shows good performance in both modes, while Random Search does not. This show the consistent performance of the same optimizer when it interplays with surrogate and tabular benchmarks.

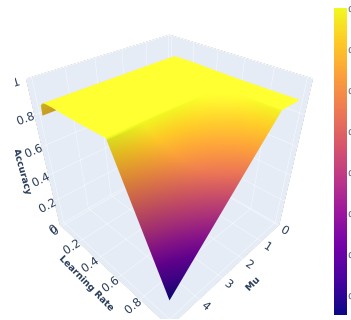

| Model | Dataset | Algo. | Train MAE | Test MAE |
|---|---|---|---|---|
| CNN | FEMNIST | FedAvg | 0.00609 | 0.00777 |
| BERT | CoLA | FedAvg | 0.04724 | 0.05454 |
| | | FedOpt | 0.02426 | 0.02959 |
| | SST2 | FedAvg | 0.02597 | 0.03227 |
| | | FedOpt | 0.02802 | 0.03166 |
| GNN | Cora | FedAvg | 0.04702 | 0.04839 |
| | | FedOpt | 0.05703 | 0.05893 |
| | CiteSeer | FedAvg | 0.01334 | 0.01381 |
| | | FedOpt | 0.01652 | 0.01717 |
| | PubMed | FedAvg | 0.04042 | 0.04148 |
| | | FedOpt | 0.04816 | 0.05699 |

**Figure 10:** Landscape with the different regularity of FedProx on Cora.

**Table 12:** The regression error of surrogate models.

### H.4.3    VARIANCE OF DIFFERENT SAMPLE RATE

As we build our tabular benchmark from FL courses executed in docker images provided by FS, we can fully reproduce all the results given the same random seed in raw mode. Other than that, to study the noise of different federated optimization, we analyze the variance of validation loss with 500 rounds under different sample rates in FEMNIST. And the mean standard deviation validation loss is {1.945e-2, 1.7e-2, 1.728e-2, 1.715e-2, 1.43e-2} with sample rate {0.2, 0.4, 0.6, 0.8, 1.0}, which shows that the higher sample rate tends to have lower variance. The reason is apparent: the lower the sampling rate, the more inconsistent the set of clients sampled during the training process leads to this error.

### H.4.4    ECDF WITH DIFFERENT HETEROGENEITY

We extend our LR benchmarks with different heterogeneity settings. As we discussed in Appendix D, we split the tabular dataset with LDA, whose $\alpha$ is in {0.1, 0.5, 0.7} (the smaller the alpha, the more the heterogeneous). We show the ECDF of the normalized regret of evaluated configurations with different $\alpha$ in Figure 11, which shows that as the $\alpha$ decreases, it is harder to find a good configuration. This phenomenon shows the necessity of tuning hyperparameters in FL with heterogeneous data.

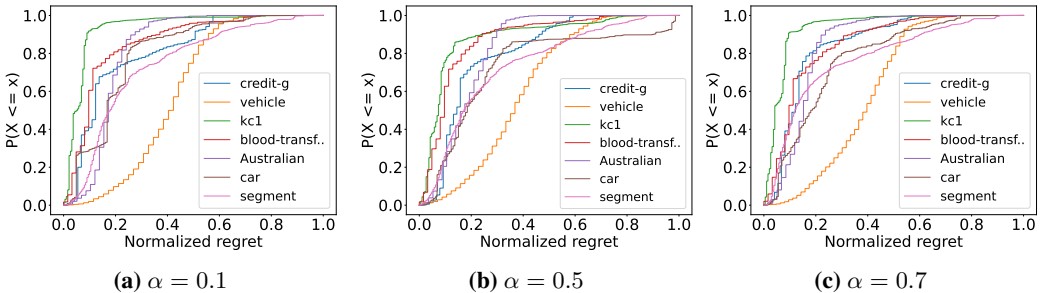

**(a)** $\alpha = 0.1$      **(b)** $\alpha = 0.5$      **(c)** $\alpha = 0.7$

**Figure 11:** Empirical Cumulative Distribution Functions with different heterogeneity in LR benchmark.

# I   MORE RESULTS

In this section, we show the detailed experimental results of the optimizers on FEDHPO-BENCH benchmarks under different modes. We first report the averaged best-seen validation loss, from which the mean rank over time for all optimizers can be deduced. Due to time and computing resource constraints, we do not have a complete experimental result of the raw mode, which we will supplement as soon as possible.

## I.1   TABULAR MODE

Following Section 4.1, we show the overall mean rank overtime on all FedHPO problems with FedOpt, whose pattern is similar to that of FedAvg in Figure 4. Then, we report the final results with FedAvg and FedOpt in Table 13 and 14, respectively. Finally, we report the mean rank over time in Figure 13-21. Due to time and computing resource constraints, the results on CNN benchmark are incomplete (lacking that with FedOpt), which we will supplement as soon as possible.

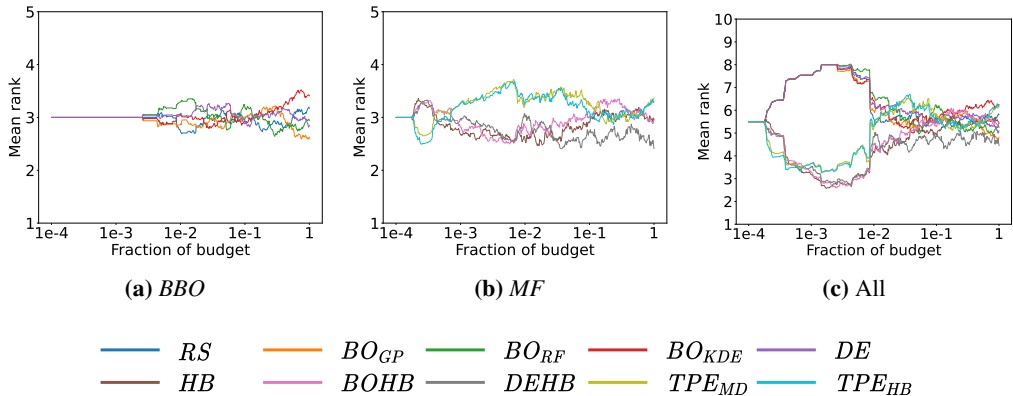

**(a)** *BBO*      **(b)** *MF*      **(c)** All

**Figure 12:** Mean rank over time on all FedHPO problems (with FedOpt).

**Table 13:** Final results of the optimizers on tabular mode with FedAvg (lower is better).

| benchmark | RS | $BO_{GP}$ | $BO_{RF}$ | $BO_{KDE}$ | DE | HB | BOHB | DEHB | $TPE_{MD}$ | $TPE_{HB}$ |
|---|---|---|---|---|---|---|---|---|---|---|
| $CNN_{FEMNIST}$ | 0.4969±0.0054 | 0.4879±0.0051 | 0.4885±0.0065 | 0.5004±0.0068 | 0.4928±0.0054 | 0.4926±0.0052 | 0.4945±0.0059 | 0.498±0.0061 | 0.5163±0.0028 | 0.5148±0.0047 |
| $BERT_{SST-2}$ | 0.435±0.0142 | 0.4276±0.0082 | 0.4294±0.0071 | 0.4334±0.0081 | 0.437±0.0052 | 0.4311±0.0151 | 0.4504±0.0441 | 0.4319±0.0251 | 0.4341±0.0044 | 0.4251±0.0076 |
| $BERT_{CoLA}$ | 0.6151±0.0014 | 0.6148±0.0014 | 0.6141±0.0016 | 0.6133±0.0022 | 0.6143±0.0006 | 0.6143±0.0016 | 0.6168±0.0025 | 0.6178±0.0025 | 0.6158±0.0014 | 0.6146±0.0018 |
| $GNN_{Cora}$ | 0.3265±0.0042 | 0.3258±0.0062 | 0.326±0.0063 | 0.3347±0.0078 | 0.3267±0.0066 | 0.3324±0.0136 | 0.3288±0.0030 | 0.3225±0.0039 | 0.3241±0.0014 | 0.3249±0.0020 |
| $GNN_{CiteSeer}$ | 0.6469±0.0052 | 0.6442±0.0046 | 0.6499±0.0069 | 0.6442±0.0089 | 0.6453±0.0061 | 0.6387±0.0077 | 0.6425±0.0054 | 0.6452±0.0030 | 0.6324±0.0070 | 0.6371±0.0051 |
| $GNN_{PubMed}$ | 0.5262±0.0167 | 0.5146±0.0136 | 0.5169±0.0193 | 0.5311±0.0110 | 0.5001±0.0082 | 0.5006±0.0144 | 0.5194±0.0212 | 0.4934±0.0010 | 0.506±0.0179 | 0.5044±0.0150 |
| $LR_{31}$ | 0.6821±0.1299 | 0.6308±0.0292 | 0.6382±0.0435 | 0.6385±0.0459 | 0.667±0.0888 | 0.6492±0.0187 | 0.6461±0.0472 | 0.6145±0.0242 | 0.7228±0.0427 | 0.758±0.0460 |
| $LR_{53}$ | 1.6297±0.1628 | 1.7288±0.2306 | 1.6116±0.2017 | 1.7142±0.1663 | 1.6062±0.1487 | 1.5765±0.1416 | 1.5634±0.1993 | 1.4755±0.1126 | 1.5506±0.0010 | 1.5506±0.0010 |
| $LR_{3917}$ | 1.8892±0.2647 | 1.7561±0.2538 | 1.7186±0.3562 | 2.4271±1.1596 | 1.7519±0.6093 | 3.948±2.5432 | 1.6384±0.1849 | 3.1183±2.9336 | 2.1344±1.0268 | 2.6576±1.1446 |
| $LR_{10101}$ | 0.548±0.0002 | 0.5483±0.0002 | 0.5482±0.0003 | 0.5487±0.0008 | 0.5481±0.0002 | 0.5504±0.0049 | 0.5505±0.0047 | 0.5516±0.0064 | 0.5483±0.0009 | 0.5487±0.0017 |
| $LR_{146818}$ | 0.5294±0.0006 | 0.5291±0.0002 | 0.5295±0.0006 | 0.5289±0.0004 | 0.5291±0.0007 | 0.5292±0.0008 | 0.529±0.0004 | 0.5293±0.0002 | 0.5328±0.0055 | 0.5387±0.0186 |
| $LR_{146821}$ | 0.4733±0.0025 | 0.464±0.0068 | 0.4722±0.0123 | 0.4843±0.0205 | 0.4971±0.0312 | 0.4678±0.0109 | 0.4747±0.0127 | 0.4707±0.0095 | 0.4792±0.0083 | 0.4688±0.0086 |
| $LR_{146822}$ | 0.4581±0.0202 | 0.4481±0.0102 | 0.4505±0.0182 | 0.4731±0.0197 | 0.4587±0.0118 | 0.4478±0.0122 | 0.4446±0.0066 | 0.4304±0.0071 | 0.4376±0.0071 | 0.4419±0.0089 |
| $MLP_{31}$ | 0.5899±0.0032 | 0.5891±0.0052 | 0.5808±0.0094 | 0.5904±0.0035 | 0.5925±0.0008 | 0.5921±0.0017 | 0.5929±0.0001 | 0.593±0.0001 | 0.593±0.0000 | 0.593±0.0000 |
| $MLP_{53}$ | 0.7795±0.0156 | 0.7373±0.0186 | 0.7849±0.0215 | 0.8215±0.1220 | 0.8068±0.0752 | 0.769±0.0226 | 0.7577±0.0222 | 0.8173±0.1407 | 0.9491±0.0951 | 1.0567±0.0158 |
| $MLP_{3917}$ | 0.3863±0.0099 | 0.3937±0.0094 | 0.3858±0.0105 | 0.3958±0.0088 | 0.383±0.0074 | 0.3895±0.0049 | 0.3911±0.0079 | 0.4084±0.0407 | 0.3979±0.0035 | 0.3988±0.0036 |
| $MLP_{10101}$ | 0.4054±0.0113 | 0.4217±0.0065 | 0.4361±0.0124 | 0.4162±0.0154 | 0.418±0.083 | 0.4137±0.0109 | 0.4152±0.0142 | 0.4102±0.0109 | 0.4522±0.0491 | 0.4352±0.0407 |
| $MLP_{146818}$ | 0.5089±0.0092 | 0.4997±0.0072 | 0.5125±0.0076 | 0.5112±0.0049 | 0.5138±0.0107 | 0.5009±0.0043 | 0.5199±0.0118 | 0.5039±0.0060 | 0.5392±0.0129 | 0.54±0.0197 |
| $MLP_{146821}$ | 0.184±0.0187 | 0.1251±0.0167 | 0.155±0.0183 | 0.1769±0.0410 | 0.1851±0.0236 | 0.1561±0.0279 | 0.1683±0.0291 | 0.1572±0.0305 | 0.1654±0.0422 | 0.1761±0.0409 |
| $MLP_{146822}$ | 0.2839±0.0259 | 0.2892±0.0363 | 0.317±0.0147 | 0.3586±0.0754 | 0.2928±0.0325 | 0.2927±0.0233 | 0.2823±0.0445 | 0.2549±0.0176 | 0.2745±0.0334 | 0.2755±0.0221 |

**Table 14:** Final results of the optimizers on tabular mode with FedOpt (lower is better).

| benchmark | RS | $BO_{GP}$ | $BO_{RF}$ | $BO_{KDE}$ | DE | HB | BOHB | DEHB | $TPE_{MD}$ | $TPE_{HB}$ |
|---|---|---|---|---|---|---|---|---|---|---|
| $BERT_{SST-2}$ | 0.441±0.0049 | 0.4325±0.0125 | 0.4301±0.0087 | 0.4463±0.0093 | 0.4351±0.0185 | 0.4403±0.0064 | 0.4295±0.0066 | 0.4285±0.0068 | 0.4293±0.0106 | 0.4332±0.0122 |
| $BERT_{CoLA}$ | 0.616±0.0008 | 0.616±0.0011 | 0.6141±0.0022 | 0.6137±0.0025 | 0.6159±0.0005 | 0.6154±0.0013 | 0.6157±0.0018 | 0.6176±0.0004 | 0.6172±0.0005 | 0.6168±0.0004 |
| $GNN_{Cora}$ | 0.3264±0.0027 | 0.3235±0.0004 | 0.3268±0.0032 | 0.3322±0.0101 | 0.3256±0.0009 | 0.3245±0.0014 | 0.3347±0.0121 | 0.3254±0.0008 | 0.3405±0.0129 | 0.3361±0.0187 |
| $GNN_{CiteSeer}$ | 0.6483±0.0028 | 0.6517±0.0053 | 0.6497±0.0050 | 0.6535±0.0072 | 0.6458±0.0028 | 0.6442±0.0034 | 0.6543±0.0112 | 0.6463±0.0029 | 0.6488±0.0008 | 0.6495±0.0007 |
| $GNN_{PubMed}$ | 0.4777±0.0118 | 0.4426±0.0132 | 0.4718±0.0204 | 0.4943±0.0359 | 0.4318±0.0001 | 0.4559±0.0135 | 0.4699±0.0248 | 0.4318±0.0001 | 0.4368±0.0098 | 0.4402±0.0167 |
| $LR_{31}$ | 0.7358±0.0937 | 0.6831±0.0198 | 0.6849±0.0523 | 0.8152±0.1180 | 0.7085±0.0660 | 0.6772±0.0527 | 0.6877±0.0561 | 0.6385±0.0498 | 0.8652±0.0851 | 0.7044±0.0403 |
| $LR_{53}$ | 1.7838±0.2698 | 1.5609±0.1957 | 1.5241±0.0547 | 1.5116±0.0437 | 1.6208±0.3794 | 1.6045±0.2433 | 1.7236±0.4056 | 1.3488±0.1343 | 1.6654±0.2338 | 1.7978±0.2937 |
| $LR_{3917}$ | 2.254±0.5724 | 2.0316±0.5246 | 2.3952±0.7949 | 1.9788±0.5290 | 2.6261±0.5535 | 2.3472±1.2238 | 2.5452±0.5266 | 2.3144±0.8685 | 3.2131±2.2754 | 2.0291±0.3674 |
| $LR_{10101}$ | 0.5533±0.0078 | 0.55±0.0036 | 0.5505±0.0032 | 0.5509±0.0032 | 0.549±0.0012 | 0.5504±0.0029 | 0.5476±0.0017 | 0.5522±0.0056 | 0.5612±0.0201 | 0.8567±0.6019 |
| $LR_{146818}$ | 0.511±0.0099 | 0.506±0.0103 | 0.5034±0.0097 | 0.5133±0.0078 | 0.5007±0.0021 | 0.5032±0.0054 | 0.5086±0.0067 | 0.4974±0.0030 | 0.4983±0.0049 | 0.5104±0.0157 |
| $LR_{146821}$ | 0.4017±0.0272 | 0.3599±0.0148 | 0.4121±0.0188 | 0.4134±0.0364 | 0.4079±0.0242 | 0.395±0.0228 | 0.398±0.0448 | 0.3902±0.0300 | 0.4447±0.0447 | 0.4625±0.0735 |
| $LR_{146822}$ | 0.3972±0.0060 | 0.4211±0.0236 | 0.4037±0.0191 | 0.4442±0.0261 | 0.4075±0.0127 | 0.4131±0.0215 | 0.4008±0.0085 | 0.3916±0.0086 | 0.3878±0.0074 | 0.3871±0.0043 |
| $MLP_{31}$ | 0.5912±0.0012 | 0.5914±0.0024 | 0.5912±0.0012 | 0.5912±0.0023 | 0.5918±0.0011 | 0.5923±0.0007 | 0.5921±0.0007 | 0.5911±0.0023 | 0.5921±0.0006 | 0.5921±0.0006 |
| $MLP_{53}$ | 0.9096±0.0690 | 0.8166±0.0890 | 0.8111±0.0998 | 0.8872±0.1465 | 0.8546±0.1223 | 1.0163±0.0781 | 0.8565±0.0574 | 0.9849±0.1238 | 1.1276±0.0394 | 1.0952±0.0293 |
| $MLP_{3917}$ | 0.3798±0.0126 | 0.3937±0.0086 | 0.3862±0.0075 | 0.3871±0.0110 | 0.3867±0.0086 | 0.4109±0.0402 | 0.4262±0.0687 | 0.3812±0.0125 | 0.4003±0.0000 | 0.4003±0.0001 |
| $MLP_{10101}$ | 0.4219±0.0168 | 0.4141±0.0056 | 0.4197±0.0138 | 0.4111±0.0078 | 0.4303±0.0369 | 0.4145±0.0106 | 0.4256±0.0286 | 0.4215±0.0277 | 0.4502±0.0390 | 0.4502±0.0300 |
| $MLP_{146818}$ | 0.4943±0.0018 | 0.4913±0.0108 | 0.5022±0.0090 | 0.5023±0.0113 | 0.4884±0.0058 | 0.4995±0.0087 | 0.5046±0.0298 | 0.4921±0.0293 | 0.4978±0.0135 | 0.4861±0.0240 |
| $MLP_{146821}$ | 0.1169±0.0128 | 0.0836±0.0132 | 0.0915±0.0149 | 0.1674±0.0600 | 0.1079±0.0444 | 0.0891±0.0150 | 0.1389±0.0465 | 0.0838±0.0270 | 0.1051±0.0138 | 0.1194±0.0130 |
| $MLP_{146822}$ | 0.2963±0.0264 | 0.2914±0.0215 | 0.2705±0.0240 | 0.3025±0.0447 | 0.2779±0.0063 | 0.2759±0.0216 | 0.2621±0.0201 | 0.2549±0.0108 | 0.257±0.0020 | 0.2518±0.0055 |

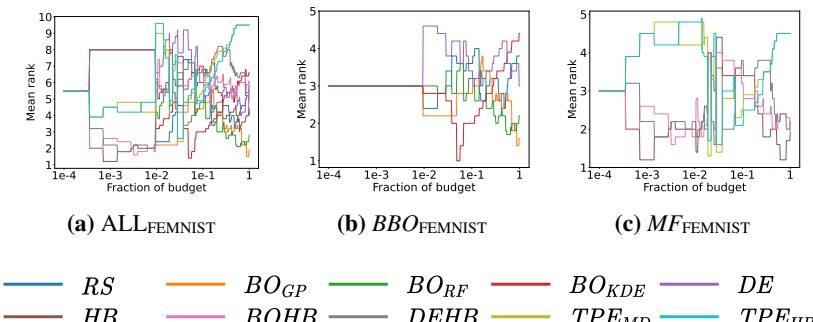

**(a)** ALL$_{\text{FEMNIST}}$     **(b)** $BBO_{\text{FEMNIST}}$     **(c)** $MF_{\text{FEMNIST}}$

**Figure 13:** Mean rank over time on CNN benchmark (FedAvg).

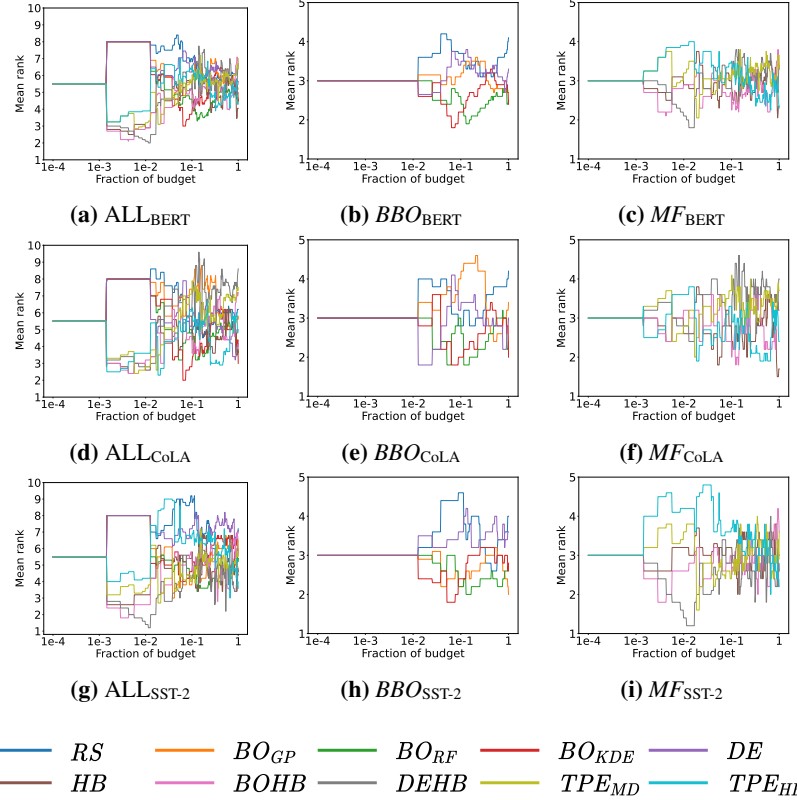

**(a)** ALL$_{\text{BERT}}$     **(b)** $BBO_{\text{BERT}}$     **(c)** $MF_{\text{BERT}}$

**(d)** ALL$_{\text{CoLA}}$     **(e)** $BBO_{\text{CoLA}}$     **(f)** $MF_{\text{CoLA}}$

**(g)** ALL$_{\text{SST-2}}$     **(h)** $BBO_{\text{SST-2}}$     **(i)** $MF_{\text{SST-2}}$

**Figure 14:** Mean rank over time on BERT benchmark (FedAvg).

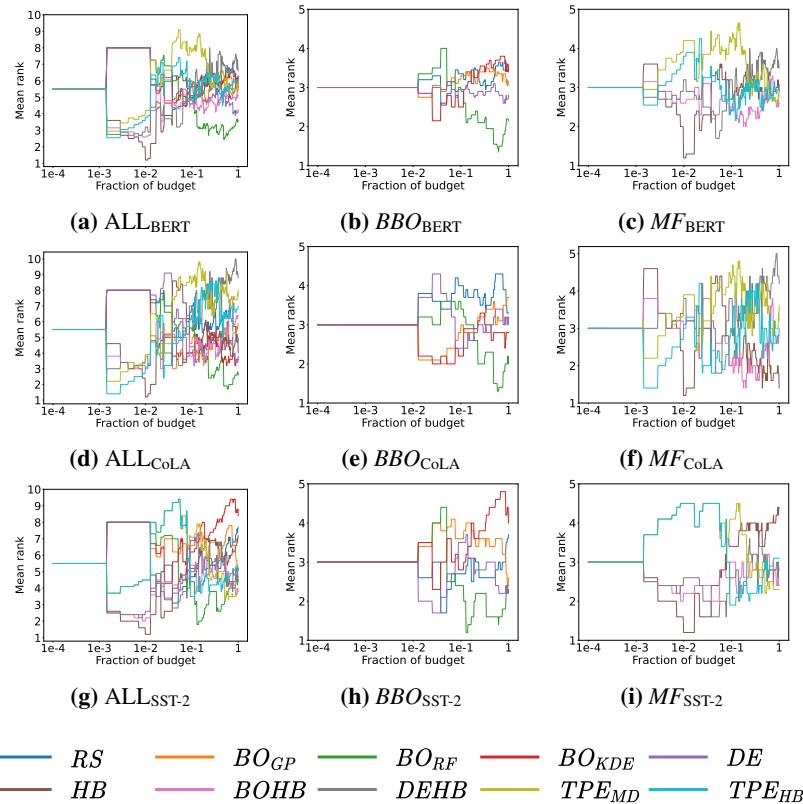

**Figure 15:** Mean rank over time on BERT benchmark (FedOpt).

## I.2 SURROGATE MODE

We report the the final results with FedAvg on FEMNIST and BERT benchmarks in Table 15. Then we present the mean rank over time of the optimizers in Figure 22 and Figure 23.

**Table 15:** Final results of the optimizers in surrogate mode (lower is better).

| benchmark | RS | $BO_{GP}$ | $BO_{RF}$ | $BO_{KDE}$ | DE | HB | BOHB | DEHB | $TPE_{MD}$ | $TPE_{HB}$ |
|---|---|---|---|---|---|---|---|---|---|---|
| CNN$_{FEMNIST}$ | 0.0508 | 0.0478 | 0.0514 | 0.0492 | 0.0503 | 0.0478 | 0.048 | 0.0469 | 0.0471 | 0.0458 |
| BERT$_{SST-2}$ | 0.4909 | 0.4908 | 0.4908 | 0.4908 | 0.4908 | 0.4908 | 0.4908 | 0.4917 | 0.4908 | 0.4908 |
| BERT$_{CoLA}$ | 0.5013 | 0.4371 | 0.4113 | 0.487 | 0.444 | 0.4621 | 0.4232 | 0.4204 | 0.3687 | 0.3955 |

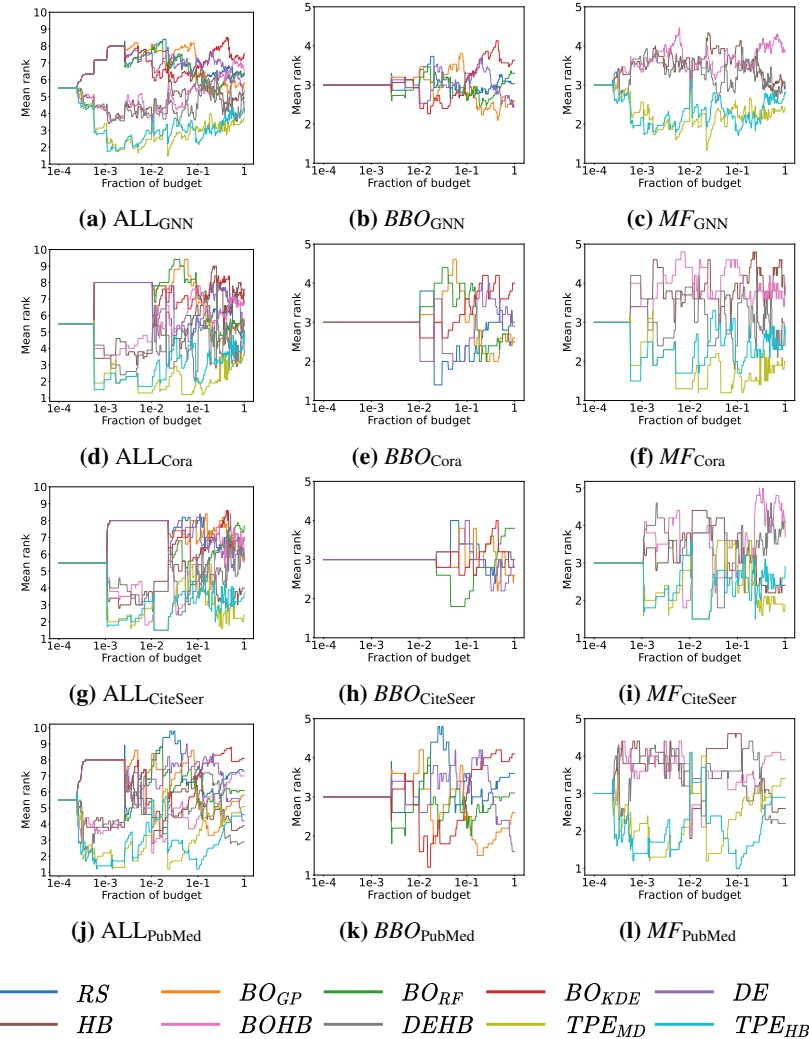

**Figure 16:** Mean rank over time on GNN benchmark (FedAvg).

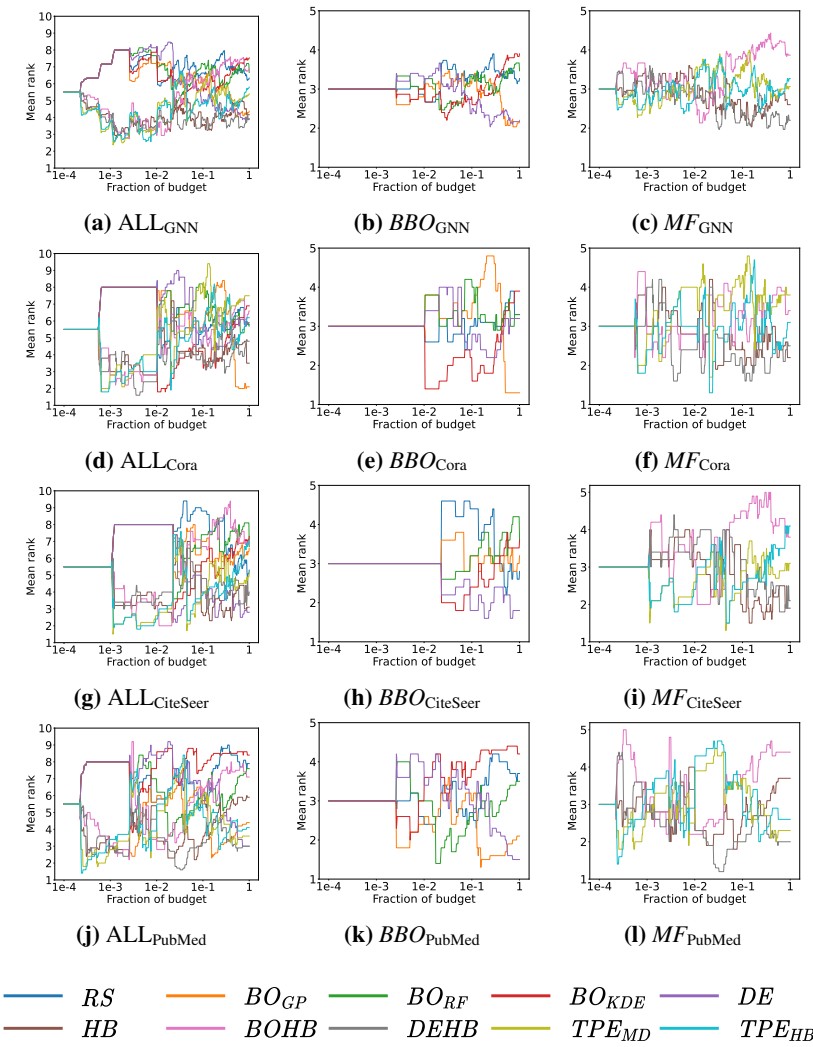

**Figure 17:** Mean rank over time on GNN benchmark (FedOpt).

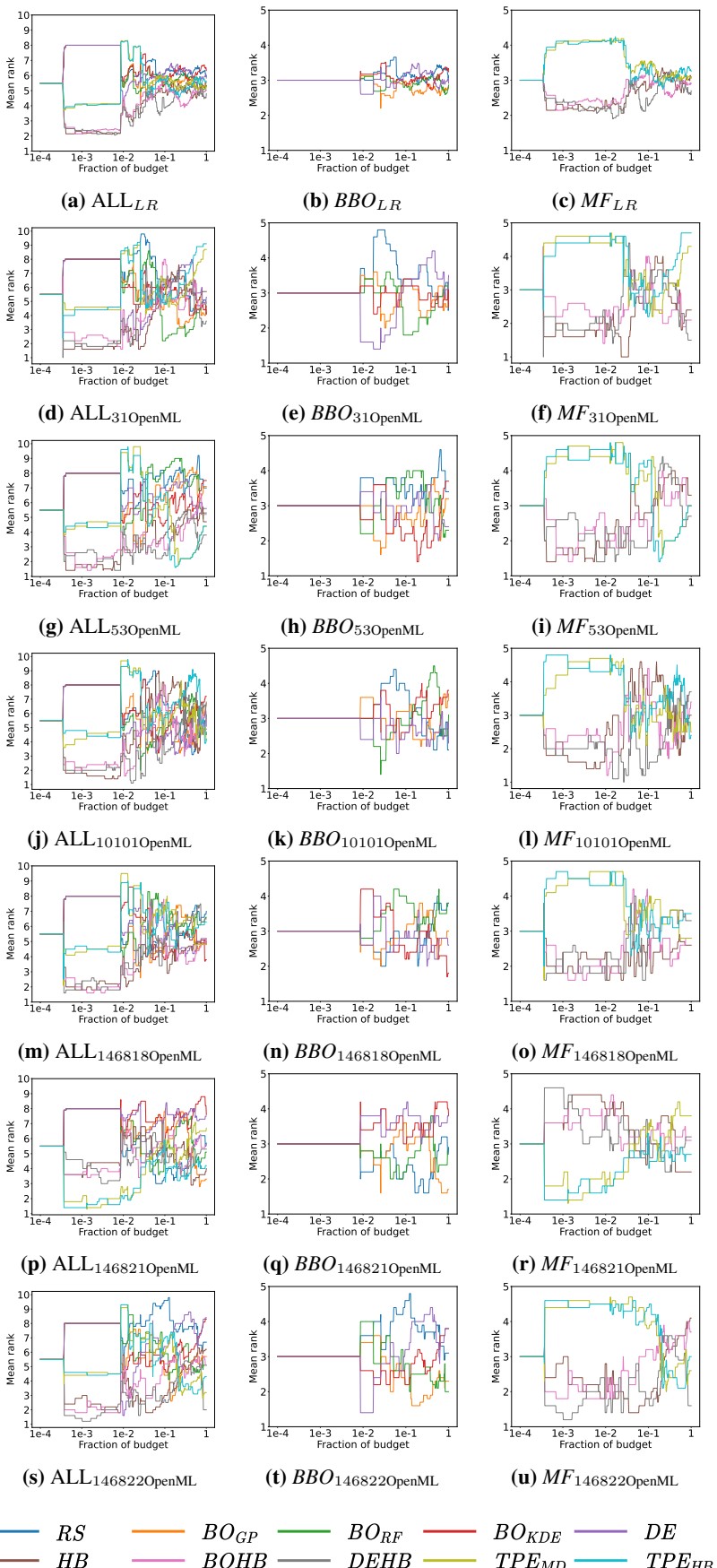

**Figure 18:** Mean rank over time on LR benchmark (FedAvg).

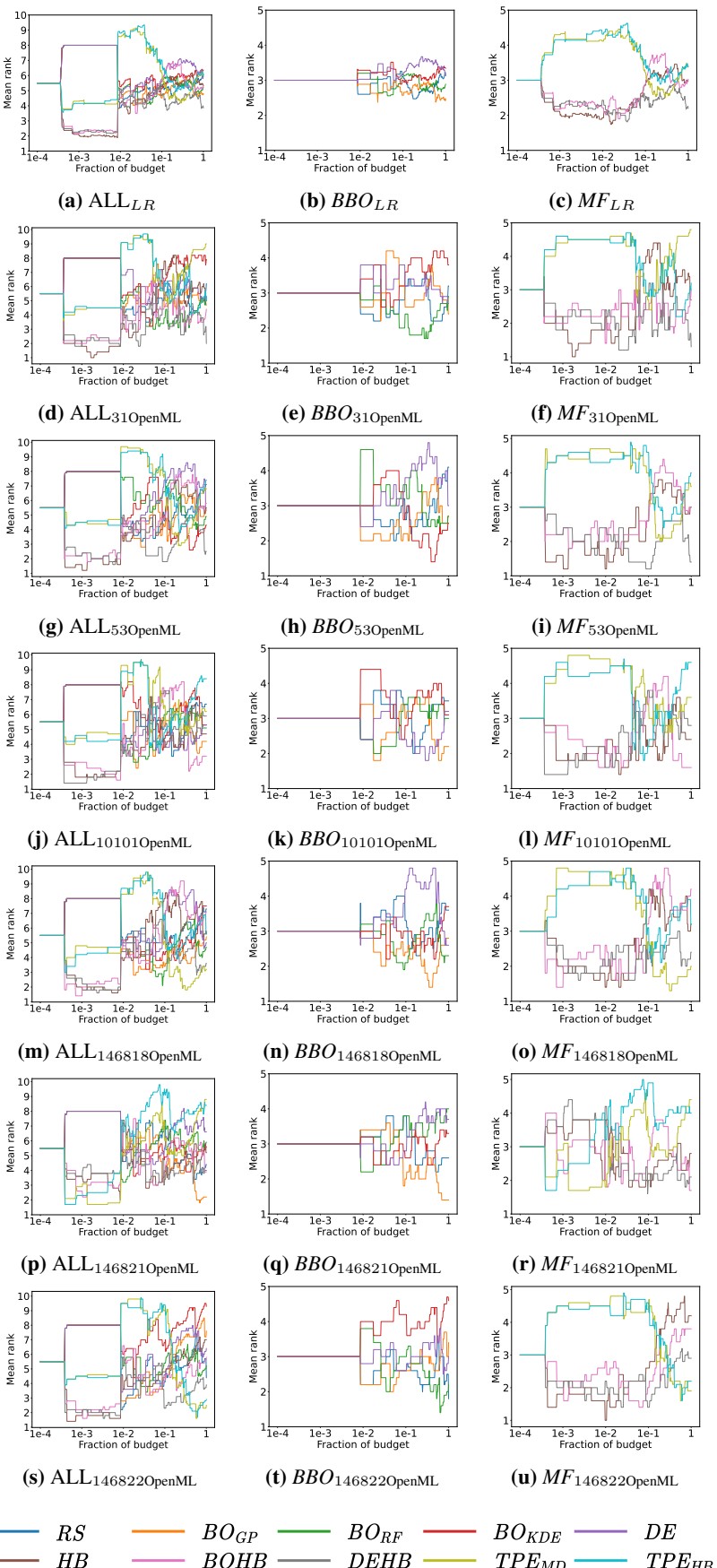

**Figure 19:** Mean rank over time on LR benchmark (FedOpt).

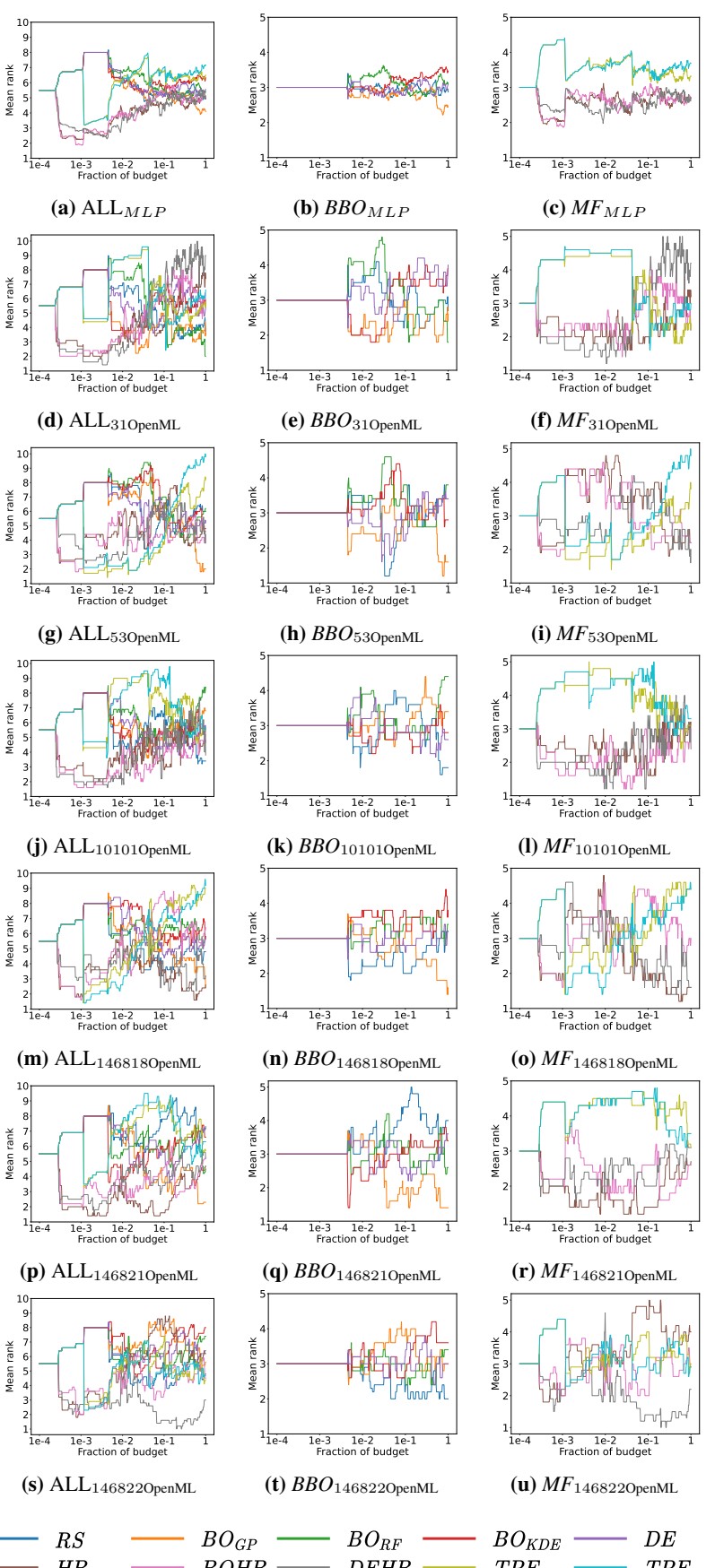

**Figure 20:** Mean rank over time on MLP benchmark (FedAvg).

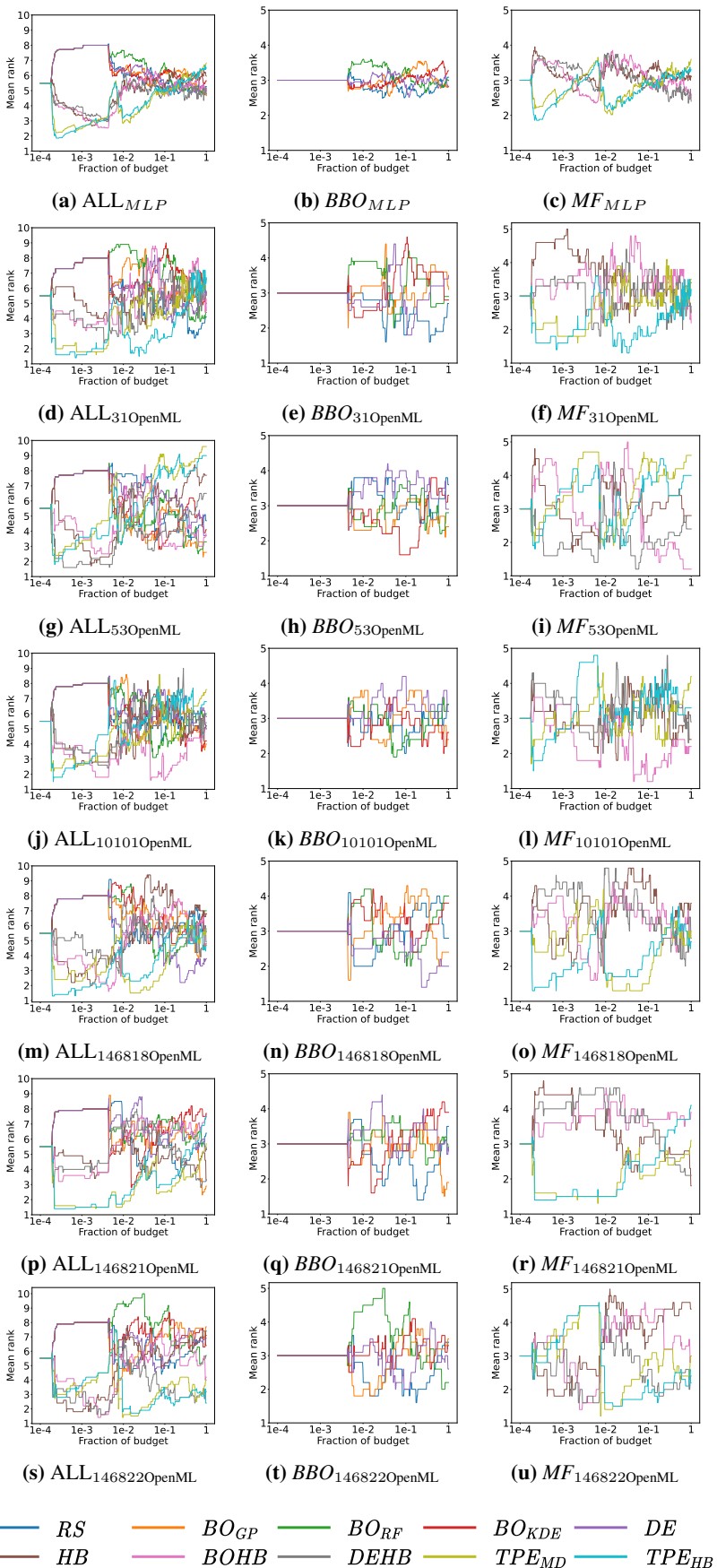

**Figure 21:** Mean rank over time on MLP benchmark (FedOpt).

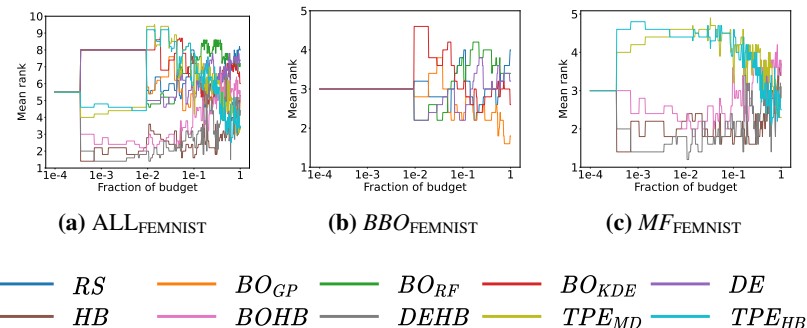

**Figure 22:** Mean rank over time on CNN benchmark under surrogate mode (FedAvg).

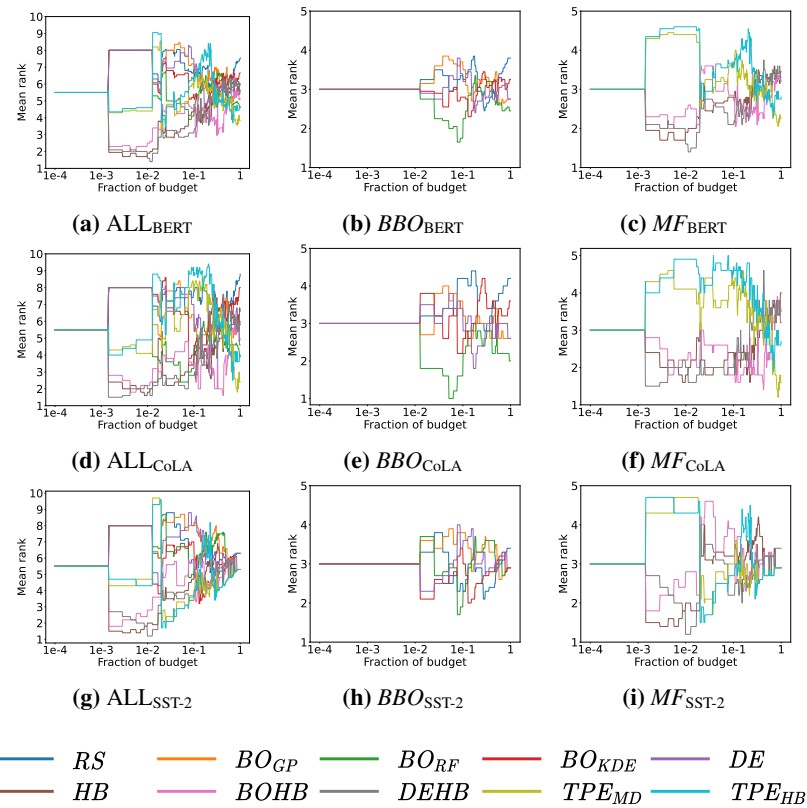

**Figure 23:** Mean rank over time on BERT benchmark under surrogate mode (FedAvg).

