# OpenReview forum: "FedHPO-Bench: A Benchmark Suite for Federated Hyperparameter Optimization"
_ICLR.cc/2023/Conference — Submitted to ICLR 2023_

### Official Review · Reviewer_BCqK · 2022-10-20

**Confidence:** 4
**Correctness:** 2
**Technical Novelty And Significance:** 3
**Empirical Novelty And Significance:** 2
**Recommendation:** 5

**Clarity, Quality, Novelty And Reproducibility:**

### Quality
As the list of weaknesses shows, the quality of this work is not convincing. I believe in the general idea, but the overall quality of the paper itself is not at a stage where it should be published.
### Clarity
Language-wise, clarity is great. I had no problem understanding any point made. However, considering the overall story of the paper, clarity is not great. As written earlier, the arguments conducted by the authors are not compelling. I suggest to completely rewrite the paper following a clearer structure to make clearer a) what the federated HPO problem is, b) why and how it is different from the standard HPO one, c) why different HPO approaches are needed, d) why a special benchmark suite is needed and e) how the benchmark looks like exactly.
### Novelty
As this seems to be the first HPO benchmark tailored to federated learning, novelty is great.
### Reproducibility
Reproducibility is, in general, guaranteed as the code of the benchmark and evaluation is released. That being said, I would like to see some more details given in the actual evaluation section of the main paper.


**Strength And Weaknesses:**

### Strengths
* As federated learning becomes more and more important in practice, considering HPO for federated learning is an important research direction. The idea of having a corresponding benchmark is in general an important contribution.
* Language-wise the paper is well-written.

### Weaknesses
* The overall structure of the paper is not very compelling. The authors spend quite some space on explaining the differences between the standard HPO and the federated HPO setting without properly defining the latter. There is a vague description of the overall process on Page 7 (directly before the experiments), which should be a) much clearer and b) much earlier in the paper to convince the reader that the two problems are truly distinct. Overall, I am missing a clear path to follow through the paper, which makes it not very convincing.
* The points made by the authors regarding the differences between the standard HPO and federated HPO problem are not convincing to me. Each difference detailed on Page 3 can very well be modeled within the realm of standard HPO:
    * Function evaluation in FL: The argument for FL algorithms being iterative is not well supported. Many standard ML models such as neural networks are also iteratively trained making FL not very different from it. However, we still tune neural networks using standard HPO methods. Similarly, the heterogeneity between FL clients is not well elaborated on. Why does this change the HPO problem in particular?
    * Hyperparameter dimensions: New hyperparameters are not a reason for a new HPO formulation and benchmark. If this was the case, we would need a new benchmark with each new ML algorithm or variant thereof. Showing the figure in Appendix H.3.1 is not very convincing as different HPO problem instantiations (e.g. with different algorithms) also can have different shapes. Moreover, the figure is rather destructive to the overall motivation for the federated HPO problem as it suggests that the optimal region is in a very large plateau, which can easily be found by random search. In fact this observation is even corroborated in the experiments later on where the authors argue that none of the standard HPO approaches shows a significant improvement. In principle, this is also not such a new insight as something similar can also be observed for HPO problems in general [7].
    * Fidelity dimensions: We can have different fidelity dimensions in standard HPO as well. Moreover, the argument conducted simply shows that some of the hyperparameters are correlated - an issure that we also have on standard HPO.
    * Privacy: While this is certainly an important property to be analyzed, it seems that this is, once again, simply controlled via a (set of) hyperparameters, which could also be modeled in the standard HPO setting. Privacy simply seems to be a second objective suggesting that federated HPO is actually a multi-objective HPO problem. Unfortunately, this is never mentioned by the authors.
    * Fairness: First of all, fairness often has a very different notion in the area of machine learning nowadays, which might be very confusing for readers of this paper. The authors should make much clearer, what they mean by privacy as was done in the corresponding references given by the authors. Second, the description here suggests that federated HPO should be multi-objective.
    * Concurrent exploration: Once again, this is also something that exists in standard HPO and is in fact quite beneficial to perform in the standard setting. For example, [2] suggests a UCB based acquisition function, which can be used in concurrent exploration as done in D-SMAC [3].
Overall this makes the description of differences not very convincing. To be clear: I am very sure that the problem features several important characteristics, which make it inefficient to treat it using standard HPO methods. But this is simply not well conveyed by the authors. Most importantly, the overall setup, i.e. how the process works (description directly before evaluation) already suggests that tuning should NOT be performed as standard in the sense that a single function evaluation consists of several rounds between the clients and the master until convergence, but that the iterative process itself should be abused. In fact, this suggests the usage of multi-fidelity approaches or, if the tuning should be performed duringthe training of the federated approaches as it is suggested in this work, using methods from dynamic algorithm configuration [4,5]. The authors should really consider completely rethinking the way they present (a) the federated learning problem and (b) its differences to the standard setting.
* It is unclear why the authors do not build upon existing benchmarks for federated learning such as [1] and why existing federated HPO methods have not been well benchmarked as claimed by the authors. Once again, this does not make a compelling point for the need for this work.
* The evaluation is not very compelling.
    * Firstly, RQ3 is raised in the main paper, but never again mentioned there. It is answered in the appendix, but this makes the main paper not self-contained. The authors should either answer the question within the main paper or not raise it there in this form in the first place, but completely defer it to the appendix.
    * Secondly, I do not see on the basis of which federated HPO problems RQ1 is answered. While Figure 1 looks interesting, the results are not well put into context. As mentioned earlier, Appendix H.3.1 suggests that the problem can be solved well by random search. Correspondingly, the result "Comparing these advanced optimizers with their baselines, only BOGP, BORF, and DE win on more than half of the problems but have no significant improvement, which is inconsistent with the non-FL setting." is not very surprising, which is not mentioned. Moreover, the second result "Meanwhile, no MF optimizers show any advantage in exploiting experience, which differs from non-FL cases." cannot be verified from my point of view. It seems that some optimizers do work better than others at earlier stages of the optimization from Fig. 4. Other than that, I am not sure how you come to that conclusion from the results shown in the paper.
    * Thirdly, the experiment to answer RQ2 is limited to only a single federated HPO task, which seems strange considering that the authors suggest a complete benchmark here, which should feature many more problems. Moreover, it is unclear which loss function is displayed in Figure 5 (validation of what loss?) making the figure meaningless as it cannot be properly interpreted.
    * Fourthly, the evaluation completely lacks error bands or standard deviation values to put the result into context.
* Important details regarding the benchmark are missing in the description. For example, what are the benchmark problems contained? How many problems does it contain? How exactly does it support the user in evaluating federated HPO algorithms? Which algorithms are already implemented? Will the benchmark be further extended by the authors?
* There exist several notational and descriptional problems in the work. For example, the HPO problem description in the introduction is not great as the in the end one is not interested in optimizing the validation performance, but rather the test performance. In fact, one is actually interested in optimizing the out of sample error, which is approximated by the test error. Similarly, several symbols of Eq. 1 are not defined: What is S_{down}, S_{up}, B_{down}, B_{up}?

## Minor remarks
* Figure 1: text "$\lambda, b$ as query" is too long and is displayed above another part of the figure
* p.6: "As a result, the learning dynamics of FL change, and thus different privacy budgets often correspond to different HPO objective functions" -> I disagree. The objective function does not change with a different privacy hyperparameter, as the objective is always to maximize performance, as far as I can tell. What might change is your additional objective, if you explicitly make privacy an objective, but I do not read anything regarding multi-dimensional HPO in the paper.
* p.6: model execution time calibration -> The authors should consider citing [6], where something similar is done to adhere to varying hardware.
* Eq. 1: Several symbols undefined -> not possible to understand the idea from the main paper here
    * What is S_{down}, S_{up}, B_{down}, B_{up}
* p. 7 last sentence: unclear - what is "#round"?

### Typos:
* p.4: "much helpful" -> very helpful
* p.7: "it needs to synchronous" -> "it needs to synchronize"

[1] He, C., Li, S., So, J., Zhang, M., Wang, H., Wang, X., Vepakomma, P., Singh, A., Qiu, H., Shen, L. and Zhao, P., 2020. Fedml: A research library and benchmark for federated machine learning. arXiv preprint arXiv:2007.13518
[2] Donald R. Jones. A taxonomy of global optimization methods based on response surfaces. Journal of Global Optimization, 21(4):345–383, 2001.
[3] Frank Hutter, Holger H. Hoos, & Kevin Leyton-Brown. Parallel algorithm configuration. In International Conference on Learning and Intelligent Optimization, pages 55–70. Springer, 2012.
[4] Biedenkapp, André, et al. "Dynamic algorithm configuration: foundation of a new meta-algorithmic framework." ECAI 2020. IOS Press, 2020. 427-434.
[5] Adriaensen, Steven, et al. "Automated Dynamic Algorithm Configuration." arXiv preprint arXiv:2205.13881 (2022).
[6] Mohr, Felix, et al. "Predicting machine learning pipeline runtimes in the context of automated machine learning." IEEE Transactions on Pattern Analysis and Machine Intelligence 43.9 (2021): 3055-3066.
[7] PUSHAK, YASHA, and HOLGER H. HOOS. "AutoML Loss Landscapes." ACM Transactions on Evolutionary Learning and Optimization (TELO) (2022).


**Summary Of The Paper:**

The authors propose a benchmark suite for hyperparameter optimization (HPO) specifically tailored to the problem of federated learning. They argue that the problem features several distinct characteristics, which are not accounted for in standard benchmarks, but should be. FedHPO-Bench was designed to alleviate this situation with the corresponding differences in mind. In order to increase the size of available federated HPO problems, the authors also present a way to generate a federated HPO problem from a standard one. In an experimental study, the authors show how standard and federated HPO approaches perform on federated HPO problems from the proposed benchmark.

**Summary Of The Review:**

Although I very much like the idea of investigating the HPO problem for federated learning, I have to suggest a rejection of the paper for the following reasons:

1. The overall story of the paper is not well written making it unclear a) what the federated HPO problem actually is and in how far it differs from the standard HPO problem, b) why existing HPO methods are not well suited, c) why this benchmark is needed and what it offers in detail.
1. The evaluation presented in the paper has several flaws, which should be fixed.

---

> ### Author Response · Authors · 2022-11-19
> **Our Response [part1]**
>
> Thanks for your time and efforts in doing this review! We have revised our submission accordingly and highlighted the changes in blue color. Regarding your concerns, we make the following clarifications:
>
> - (About not being compelling enough) We have re-written the mentioned parts to clarify the FedHPO setting in our updated submission. However, problem-solving and problem-definition are two different perspectives. The overall process on Page 7 (directly before the experiments) of our original submission is about how several recently proposed FedHPO methods proceed (i.e., problem-solving), while, in Section 2, we provide necessary preliminary, such as the definition of an HPO problem (in Section 2.1), where FedHPO also follows such a high-level formulation yet possesses its uniqueness as we have discussed in Section 2.2.

---

> > ### Author Response · Authors · 2022-11-19
> > **Our Response [part2]**
> >
> > - (About the differences between standard HPO and FedHPO) We have re-written the mentioned parts according to your suggestions to clarify the differences. We think the update submission is more informative and clear. Here we also make clarifications for our original submission as follows:
> >
> >   - Function evaluation in FL: That paragraph (i.e., the first paragraph of Section 2.2) intends to describe the process of an FL course, followed by a summarization---"the execution of an FL algorithm is essentially a distributed machine learning procedure", which immediately implies several of the discussed points of FedHPO's uniqueness. In fact, we have never attributed the uniqueness of FedHPO to the iterative nature of FL algorithms. As for the heterogeneity, we at first mentioned it as one of the differences between FL and general distributed machine learning. How it matters FedHPO is further elaborated on---"the heterogeneity among clients is likely to make them have different optimal configurations" (see the "Concurrent exploration" paragraph of Section 2.2 in our original submission).
> >
> >   - Hyperparameter dimensions: We agree that new hyperparameter dimensions should not be the motivation of a dedicated new HPO benchmark. Therefore, in the corresponding paragraph, we describe the FL-unique ones, with an emphasis on the server-side v.s. client-side categorization. As you can see from Section 4.2, such a categorization requires methods (e.g., FedEx) that instantiate the idea of concurrent exploration to adopt some traditional HPO method as a wrapper for optimizing the server-side hyperparameters. As for the figure in Appendix H.3.1, it is empirical evidence for the correlations between FL-unique hyperparameters and the general machine learning hyperparameters, which also implies the possibility of difference between FL and non-FL HPO problems. It is unreasonable to justify the existence of difference by empirical observation, and we have never done that in our discussions. Besides, as we describe in the caption of the figure, the landscape is depicted on the general machine learning-related hyperparameters (i.e., learning rate and batch size), where the FL-related hyperparameters have been chosen to be the most competitive configuration. To solve a FedHPO problem, these dimensions also need to be optimized, where the landscape might not be the case that prefers random search.
> >
> >   - Fidelity dimensions: We agree with your comments. However, in the corresponding paragraph, we emphasize FL-unique fidelity dimension *client\_sample\_rate*, which doesn't exist in the standard HPO that considers a centralized machine learning scenario. Furthermore, when discussing fidelity dimension(s), standard HPO focuses on the trade-off between evaluation precision and efficiency. In contrast, we discuss and empirically study the adjustment of two fidelity dimensions (i.e., *\#round* and *client\_sample\_rate*) to achieve more economical trade-offs regarding a given system condition.
> >
> >   - Privacy and Fairness: We agree that taking privacy and/or fairness concerns into account can be modeled as a multi-objective optimization problem. In our original submission, we handle such a multi-objective problem by regarding the fairness metric as a regularizer to the original objective (see the third paragraph of Section 3.2). We further emphasize such a point as one of the uniqueness of FedHPO in our updated submission.
> >
> >   - Concurrent exploration: In our original submission, we describe "concurrent exploration" as "trying out different client-side configurations in each round and updating a policy according to the feedback from all these clients". According to the context, each "round" belongs to one specific FL course, or say, one specific HPO trial from the black-box function perspective. Without a distributed machine learning scenario, such an idea cannot be instantiated, and thus the counterpart in standard HPO means the concurrency in another granularity/aspect. We agree that the process in Figure 3 is helpful for describing "concurrent exploration" in FedHPO, yet it is still necessary to keep in mind that this is a strategy for solving the FedHPO problem, which does not change the definition of the FedHPO problem. We have clarified the differences between FedHPO methods and these mentioned related works in our updated submission.

---

> > > ### Author Response · Authors · 2022-11-19
> > > **Our Response [part3]**
> > >
> > > - (About existing FL benchmarks) At the beginning of Section 3, we stated that FedHPO-Bench is built upon FederatedScope [1, 2], a comprehensive and extensible FL framework. Thus, it is confusing to ask why we do not adopt existing FL benchmarks, especially mentioning another instance (i.e., FedML). As for benchmarking FedHPO methods, we have discussed related FedHPO methods in Appendix B, from which readers can find that they often conduct experiments with respective problems and implementations. As an academic paper, we should not judge the motivations behind this phenomenon. Thus, we discuss the reason for this phenomenon from a technical perspective---"FedHPO methods, such as FedEx, are coupled with the FL course" (quoted from Section 3.3 of our original submission). We have added more such an explanation in Section 2.2 of our updated submission.
> > >
> > > - (About evaluation)
> > >
> > >   - In our updated submission, we completely defer the discussion of RQ3 to Appendix according to your suggestion.
> > >   - As described in Section 4.1, we largely follow the experiment made by HPO-Bench [3] to answer RQ1. As for "which FedHPO problems RQ1 is answered", we have told readers that "we apply these optimizers to solve the cross-silo FedHPO problems summarized in Table 1". We have not judged whether the experimental results are surprising. Instead, as the first FedHPO benchmarking work, we intend to validate the usability of FedHPO-Bench and gain insights into FedHPO. Whether the conclusions are consistent with that in non-FL cases is judged w.r.t. the results of HPO-Bench's experiments, where the HPO problems correspond to traditional centralized machine learning scenarios.
> > >   - On FEMNIST, an image classification task is considered. Hence, the reported validation loss is the cross-entropy loss on the validation set. We have added the description of reported metrics in our updated submission, and more experiments on our FedHPO problems are being conducted, which will be added in the next version of our paper.
> > >   - Following HPOBench [3], the error bands were omitted to make the curves look clear. But we have added such information in Table 13 and Table 14 in our updated submission to address your concerns.
> > >
> > > - (About important details) In all, the mentioned questions have been answered in our original submission: Table 1 summarizes the problems FedHPO-Bench contains. Considering "raw" mode, there are \#Dataset * \#Algo = (2+2+3+7+7+1+1) * 2 = 46 problems. FedHPO algorithms can be evaluated following the scripts we provide in our repository for reproducing the experiment in Section 4.2. We have implemented FedEx, FTS, and personalized variants of FedEx in FederatedScope, as the last paragraph of Section 3.3 states. In Appendix A, we present our maintenance plan.
> > >
> > > - (About notations and descriptions) Thanks for your suggestions! We admit that the ultimate goal of HPO is optimizing the generalization performance rather than the validation performance. Our original definition focuses on the optimization procedure, and we have clarified this difference in our updated submission. We have defined $S(f,\lambda)$ below Eq. 1 in our original submission, where the subscript "down" and "up" are eliminated as we thought they could be easily inferred without ambiguity. Now we have clarified them in our updated submission.
> > >
> > > - (About the minor remarks and typos) Thanks for your detailed comments! We have fixed them in our update submission.
> > >
> > > Thanks again for your comments and suggestions! We have re-written a large portion of our submission according to your suggestions, where all your mentioned HPO papers are cited and discussed now. We hope our explanations and improvements have addressed your concerns. If you have any further questions or suggestions, please let us know. Meanwhile, could you please kindly consider increasing the overall score? Looking forward to your reply!
> > >
> > > [1] Xie et al. "FederatedScope: A Flexible Federated Learning Platform for Heterogeneity". arXiv'22.
> > >
> > > [2] Wang et al. "FederatedScope-GNN: Towards a Unified, Comprehensive and Efficient Package for Federated Graph Learning". KDD’22.
> > >
> > > [3] Eggensperger et al. "HPOBench: A Collection of Reproducible Multi-Fidelity Benchmark Problems for HPO". NeurIPS'21.

---

> > > > ### Comment · Reviewer_BCqK · 2022-11-29
> > > > **Thanks for the reply**
> > > >
> > > > I would like to thank the authors for the detailed answer and the corresponding changes to the paper, which have made it a bit more compelling from my point of view. In fact, the related work papers helped me to better appreciate your paper. Nevertheless, I agree with the other reviewers that it is not fully ready and still has some weak points. Therefore, I increased my score only to borderline reject. Nevertheless, I believe that this is very worthwhile to pursue and further refine.

---

### Official Review · Reviewer_eSy6 · 2022-10-23

**Confidence:** 3
**Correctness:** 4
**Technical Novelty And Significance:** 3
**Empirical Novelty And Significance:** 2
**Recommendation:** 5

**Clarity, Quality, Novelty And Reproducibility:**

Clarity: Generally quite clear, except for a few details pointed out above.
Quality: Straight forward and useful idea. Potentially will have great practical impact. However, the empirical study is not quite relevant.
Novelty: As far as I know this is the first step in this direction. The unification is interesting, but needs to be better demonstrated.
Reproducibility: The authors released a github repo. (minor note -- I would recommend separating the appendix from the main submission to follow the ICLR guideline). I'm convinced that it can be reproduced.

**Strength And Weaknesses:**

Strength: The paper is well motivated. The proposed software has great practical values and is potentially impactful in the growing field of Fed Learning. While I believe there is much more to be done, the current implementation might be sufficient as a prototype.

Weakness: My main concern is that the empirical studies are not very well designed. The authors mentioned that there are three desiderata for a good benchmark suite. The experiments should focus on demonstrating these three desiderata. The insights described in Sec 4 are good to know, but not entirely relevant. My specific comments regarding this are as follows:

-- Comprehensiveness: In my opinion, this is the most well demonstrated desiderata. The authors show in Fig. 2 a wide range of hyperparameters-performance distributions, obtained over different categories of model types. But these are confined to the vanilla FL setting. It would be much better if the authors can demonstrate on other interesting settings, such as increasing/decreasing the heterogeneity of the federated tasks. For example, I would expect that increasing the heterogeneity will make the ecdf curves of most datasets be more like that of the PubMed dataset (i.e., harder to find a good configuration that works well on all clients). Many existing papers on non-IID FL have done this study before, so I believe it would be useful if this feature is implemented as an environment parameter that users have control over.

-- Flexibility: None of the experiments really demonstrate how easy it is to use the proposed software to generate interesting experiments. Varying the heterogeneity (as pointed out above) would be a great feature to have (and to conduct demonstration for). In addition, I also find it unintuitive to customize the objective function (to reflect different privacy budget/fairness level -- Sec 3.2). Personally, I think it is better to separately implement DP and fairness metrics and let the users decide which objective function to use. As the system is designed to be extensible, the dev team can incorporate implementations of privacy and fairness preserving FL objectives later.

-- Extensibility: This is arguably harder to demonstrate, and I think the description in Sec 3.3 did a fairly good job to convince me that the bench can be extended in principle. My concern is that while three concurrent exploration methods were implemented (FedEx, FTS, personalized FedEx), it seems that only FedEx was demonstrated in the experiments. In addition, does the unification result in other meaningful concurrent exploration methods by varying the sync, update & aggr operations? I would recommend the authors trying several sensible configurations and show that they yield decent results (not necessarily better) to confirm that the unification is indeed useful. That will also demonstrate the extensibility of the bench. Otherwise, it seems pointless to unify these methods.

Other concerns:

-- The authors should make clear the distinction between HPO and FedHPO in the problem setting (Sec 2.1 or beginning of 2.2). Is the goal learning one HP setting for all clients or a personalized setting for each client?
-- I do not get the discussion about the fidelity dimensions (Sec 2.2). I understand what they are and they seem like parameters of the federated environment, which FedHPO-bench should give users control over (i.e., flexibility). I also do get that it is interesting to demonstrate the effect of varying these parameters (which is done in Appendix G), but it is unclear from the description whether FedHPO-bench does have this customization feature? If so, the authors should explicitly state it.


**Summary Of The Paper:**

This paper proposes a new benchmark suite for federated hyperparameter optimization (FedHPO), called FedHPO-bench. FedHPO-bench focuses on three main desiderata, which are: (1) comprehensiveness -- which is the diversity of tasks; (2) flexibility -- which is the customizability of the federated environment; and (3) extensibility -- which is the ease of evolving the benchmark suite. The paper demonstrates the usefulness of the proposed benchmark suite on a substantial collections of experiments and through which reveals interesting insights about FedHPO.

**Summary Of The Review:**

I recommend a marginally below acceptance for now, because I'm not convinced by the empirical demonstration. I hope that the authors can provide me with good reasons to upgrade my score, because the proposed bench seems like a fundamental contribution if executed well.

---

> ### Author Response · Authors · 2022-11-19
> **Our Response**
>
> # Reviewer eSy6
>
> Thanks for your time and efforts in doing this review! We have revised our submission accordingly and highlighted the changes in blue color. Regarding your concerns, we make the following clarifications:
>
> - (About experiment design) Thanks for your insightful comments and suggestions. We thought these desiderata can be achieved by our proposed design and readily confirmed by the corresponding functionalities FedHPO-Bench provides. Thus, we turn to answer those research questions in our experiments, where these desiderata are implicitly and partially validated.
>
>   - To validate *comprehensiveness*, we conduct experiments according to your insightful suggestions. Specifically, we consider different $\alpha$ for latent Dirichlet allocation (LDA) to simulate different levels of label imbalance situations among the clients. Many training courses have been executed to create tabular benchmarks in these situations respectively so that we can draw the ECDF curves.  As shown in Appendix H4.4 of our updated submission, we present the ECDF curves with $\alpha$ in {0.1, 0.5, 0.7}, which shows that as the $\alpha$ decreases, it is harder to find a good configuration. This phenomenon implies the necessity of tuning hyperparameters in FL with heterogeneous data. It is worth noticing that, in the "raw mode", we have allowed users to control the levels of Non-IIDness.
>
>   - Towards *flexibility*, we first make a clarification here. Conceptually, privacy and fairness concerns can be taken into account at two different levels. At the level of an FL training course, HPO methods/users are allowed to specify their privacy-related choices (e.g., which DP algorithm to use and how much privacy budget is allocated) and fairness-related choices (e.g., which fairness metric to be considered in this course). As our benchmark is built upon FederatedScope, a comprehensive and extensible FL framework, many off-the-shelf ingredients exist. At the level of an HPO task, we have tailored each tabular&surrogate benchmark to various kinds of DP choices, which essentially means a collection of HPO tasks differing only in their DP choices while other properties are kept consistent. As for fairness, FedHPO-Bench monitors client-wise performances ([related code snippet](https://github.com/FedHPO-Bench/FedHPO-Bench-ICLR23/blob/995a03d872a5e2fc28a06880239d703b5298696d/fedhpobench/utils/tabular_dataloader.py#L71)) and exposes the argument to users for specifying a user-defined fairness metric ([related code snippet](https://github.com/FedHPO-Bench/FedHPO-Bench-ICLR23/blob/995a03d872a5e2fc28a06880239d703b5298696d/fedhpobench/benchmarks/tabular_benchmark.py#L60)).
>
>   - Thanks for your constructive suggestions! We are running other FedHPO methods that also instantiate the "concurrent exploration" idea. The results and analysis will be provided in the next version of our paper.
>
> - (About problem definition) We are glad to see that you have noticed our thoughts about personalized FedHPO (e.g., the client-specific black-box function in Figure 2 of our original submission). As this leads to extremely high-dimensional search space, where, to the best of our knowledge, no existing research work has addressed such a challenge, we consider the same formulation of FedHPO as traditional HPO as Sec. 2.1 presents, while we also highlight "personalization" as uniqueness of FedHPO in Sec. 2.2 of our updated submission. Besides, we add a new personalized FedHPO problem where clients have heterogeneous tasks, and we prepare tabular benchmark for this personalized FedHPO problem to promote the study of personalized FedHPO (see Table 1 of our updated submission).
>
> - (About the fidelity dimensions) Sorry for making it not clear enough. Your understanding is correct. FedHPO-Bench allows users to specify *client\_sample\_rate* so that, when considering different system conditions (through actual deployment or adjusting our provided system model), HPO methods/users can choose the suitable fidelity that leads to a better trade-off between accuracy and efficiency.
>
> Thanks again for your comments and suggestions! We hope our explanations and improvements have addressed your concerns. If you have any further questions or suggestions, please let us know. Meanwhile, could you please kindly consider increasing the overall score? Looking forward to your reply!

---

> > ### Comment · Reviewer_eSy6 · 2022-11-24
> > **Thanks for the detailed reply**
> >
> > I thank the authors for taking the time to answer my concerns. It seems that at the moment, the flexibility of the proposed FedHPO-bench is largely inherited from FedScope, whereas its extensibility is not very clearly demonstrated. As such, the empirical demonstration has not fully convinced me and my score will remain unchanged. Nonetheless I do believe that this is a step in the right direction and would encourage the authors to further revise the submission.

---

### Official Review · Reviewer_QKdH · 2022-10-24

**Confidence:** 4
**Correctness:** 3
**Technical Novelty And Significance:** 2
**Empirical Novelty And Significance:** 2
**Recommendation:** 3

**Clarity, Quality, Novelty And Reproducibility:**

The paper clearly stated their contributions to the field, the setup of their experiments, and the background surrounding their research.

The paper maintained academic rigor in their experimental evaluation, setup, and the consideration of the factors specific to FL.

The originality of this work is clear, as they introduce a small, although distinct, contribution of a new HPO benchmark suite tailored for FL algorithms. That being said, this work bears an overwhelming similarity to HPO-Bench, and generally presents itself as HPO-Bench ported to FL rather than an entirely novel benchmark suite.

**Strength And Weaknesses:**

trengths: This paper clearly described the the current FL landscape.
The explanation of its extensibility, and the experimental results give a clear idea of the given justification of FedHPO-Bench as a novel research path.

Weaknesses: There were some instances throughout the paper where information was not put forth in an easily understandable format. For instance, Equation 1 provides could be summarized by saying that the total time is bounded by the summation of the slowest client’s computation, the server’s computation, and data transfer speeds. Figure 1 also introduces another issue like this, wherein the interface design is difficult to understand. Also, on page 19 (Appendix E), citing Wikipedia isn't a usable source for an academic conference due to the nature of the source of its information.


**Summary Of The Paper:**

This paper introduced a new benchmark suite for Federated HPO called FedHPO-Bench. The FedHPO-Bench is considered distinct from other HPO benchmark suites such as HPO-Bench because of the inherent requirements for FL algorithms.

The paper first identified the need for a FedHPO benchmark suite because of its distinct separation between client and server hyperparameters, and then provided some results to evaluate the performance of their benchmark suite. The analysis of the performance differences between separated client and server hyperparameters and standard HPO algorithms defined the majority of the reasoning behind the separation of HPO and FedHPO, and the corresponding need to create their benchmark suite. The authors clearly described this distinction, but did not provide convincing reasoning behind the need for their benchmark suite.
As stated in this paper, the benchmark suite is very closely related to HPO-Bench, and does not appear to provide a truly novel solution. Instead, it appears as a small addition to an already existing technology.

**Summary Of The Review:**

I would not recommend this paper for acceptance due to the fact that it does not provide a novel enough contribution to the field. Although it may definitely improve benchmarking for FedHPO, it is specific to only a single field, and it provides few novel aspects of HPO benchmark suites in general.

While I can see how there may be some need for a solution to easily benchmark FedHPO algorithms, I remain unconvinced that this is novel enough to warrant publication.

---

> ### Author Response · Authors · 2022-11-19
> **Our Response**
>
> Thanks for your time and efforts in doing this review! We have revised our submission accordingly and highlighted the changes in blue color. Regarding your concerns, we make the following clarifications:
>
> - You summarized our paper with this sentence:
>
>   > The analysis of the performance differences between separated client and server hyperparameters and standard HPO algorithms defined the majority of the reasoning behind the separation of HPO and FedHPO, and the corresponding need to create their benchmark suite.
>
>   This is one of the **five** aspects of FedHPO's uniqueness we discussed in our original submission, and we have yet to judge which aspect is the major reason of creating a FedHPO benchmark. Users may demand a FedHPO benchmark based on various reasons, which is subjective. For example, when a user wants to seek a hyperparameter configuration that can lead to satisfactory performance and acceptable fairness among FL participants, the main reason for her to adopt our benchmark instead of any existing HPO benchmark is the fairness aspect.
>
> - (About Equation 1 in our original submission) We are glad to see that you can correctly and intuitively interpret Equation 1. We appreciate such an intuitive interpretation and have added a similar summarization to our revised submission. We present Equation 1 because our system model is implemented according to it, and the "call for paper" page of this conference encourages benchmark papers to contain more details.
>
> - (About Figure 1 of our original submission (now it is Figure 2 in the updated submission)) Sorry for making you feel difficult to understand. In the left upper part of that figure, the interaction between the "HPO method" and "FedHPO-Bench" describes the interface conceptually, where "FedHPO-Bench" encapsulates the implementation of the black-box function to be optimized. A conceptually similar interface has been illustrated in [2] for a similar purpose. Meanwhile, the code example in that figure further exemplifies the design of our interface. A similar code snippet can be found in Figure 2 of [3].
>
> - (About similarity to HPO-Bench) We have thoroughly discussed the relationship between our benchmark and HPO-Bench in Appendix B.1, where both the uniqueness and commonality are detailed. On the one hand, we prepared many HPO problems under the FL setting and established the corresponding benchmarks, which differ entirely from that in HPO-Bench (all are centralized). Meanwhile, our proposed FedHPO-Bench provides featured functionalities to consider FL-related concerns, such as privacy and fairness. Besides, we have implemented SOTA FedHPO methods, such as FedEx and FTS, into FederatedScope so that users can make fair comparisons against them. All such FL-related ingredients are beyond the scope of HPO-Bench. On the other hand, we have tried our best to keep the design of the interface as similar as that in HPO-Bench as possible, which researchers from the HPO community welcome.
>
> - (About impact and novelty) In "summary of the review", you said:
>
>   > Although it may definitely improve benchmarking for FedHPO, it is specific to only a single field, and it provides few novel aspects of HPO benchmark suites in general.
>
>   Improvements in benchmarking FedHPO are critical for FL as FL algorithms are sensitive to their hyperparameter choices, and conducting HPO for FL is much more costly than for centralized learning paradigms. Hence, such an improvement is significant to the advance of FL. Meanwhile, benchmarking a specific aspect of FL has been regarded as an essential contribution [2, 4] along with the surge of FL. Moreover, some featured functionalities of our proposed FedHPO-Bench, such as providing a customizable system model and recording client-wise performances, are also suitable for benchmarking HPO for general distributed machine learning.
>
> Thanks again for your comments and suggestions! We hope our explanations and improvements have addressed your concerns. If you have any further questions or suggestions, please let us know. Meanwhile, could you please kindly consider increasing the overall score? Looking forward to your reply!
>
> [1] Wang et al. "A field guide to federated optimization". arXiv'21.
>
> [2] Wang et al. "FederatedScope-GNN: Towards a Unified, Comprehensive and Efficient Package for Federated Graph Learning". KDD’22.
>
> [3] Eggensperger et al. "HPOBench: A Collection of Reproducible Multi-Fidelity Benchmark Problems for HPO". NeurIPS'21.
>
> [4] Chen et al. "pFL-Bench: A Comprehensive Benchmark for Personalized Federated Learning". NeurIPS’22.

---

> > ### Comment · Reviewer_QKdH · 2022-12-06
> > **Response**
> >
> > Thanks for taking the time to respond to my review.
> >
> > To start with your first point, your edits have clarified the inability for HPO methods to accurately run Fed-HPO tasks and separated each point out in section 2.2. I had initially chosen what I thought was the most important point of them to depict your main argument for Fed-HPO as a problem entirely unrepresentable through HPO tasks, and therefore the inability for HPO benchmarks to accurately represent Fed-HPO tasks.
> >
> > The five separated points clearly identify the different points you choose to make, and further clarify why each point is distinct and proves the need for your benchmark.
> >
> > One note about the use of multi-objective optimization as a need for this benchmark is that it is already implemented in HPO-Bench and therefore isn't a novel requirement for an HPO task.
> >
> > To further clarify the stance taken with my prior review on why this work does not meet the criteria for passing, my main concern with this paper was the lack of robust experimental results to prove the need for this paper.
> >
> > Specifically, your research questions should answer the question as to why we need this new benchmark. Theoretically, you have described why prior HPO tasks are unusable. For RQ1, the experiment laid out here determines the rank of different HPO problems when applied to Fed-HPO problems. Within the appendix, you show a larger table of results to depict the different performances within your experiment.
> >
> > That being said, without a better frame of reference to determine how this compares to what we would expect the performance to look like, it is difficult to grasp how your experiment is able to answer RQ1.
> >
> > For the optimizer-wise comparison, we can get a very general understanding as to how they relatively perform compared to a non-FL setting, and good evidence that each method performs statistically closer to RS than in the non-FL setting. This implies a worse performance, but does not go about describing this relationship. If we were to tie this back to the original question, the attempted resolution of this experiment is the idea that because they perform differently on Fed-HPO problems than non-FL, we need a new benchmark so that we may easily study this behavior.
> >
> > This is a weak advertisement of the need for this benchmark because when benchmarking general HPO algorithms, most researchers will not need to refer to their performance on FL tasks. These algorithms were designed for general HPO tasks, and while they perform worse on FL tasks, that is a more anecdotal piece of evidence that Fed-HPO tasks are not best solved by HPO algorithms.
> >
> > For RQ2, this research question answers the need for this benchmark by comparing the performance of non-FL approaches with FL approaches. Since the FL approaches have a higher performance, the determination put forth is that the newer benchmark is needed to easily showcase the different performances of HPO methods on FL tasks. Therefore, because the performances differ, we need to be able to benchmark this in order to show the better performances offered by FL approaches.
> >
> > Between these two research questions, we are able to identify that optimization algorithms perform differently on FL and non-FL tasks, and that FL approaches are better for FL tasks. While RQ2 partially answers the need for a Fed-HPO benchmark, this paper does not provide a holistic view to show how well the benchmark actually works (i.e. how well it benchmarks different Fed-HPO algorithms). It implies that the usage of pre-existing FL tasks offers a comprehensive analysis for any Fed-HPO algorithm, but provides no evidence towards this fact.

---

### Decision · Program_Chairs · 2023-01-20

**Decision:**

Reject

**Justification For Why Not Higher Score:**

* The current empirical validation does fully support the list of claimed desiderata.
* Missing clarification/motivation around federated HPO problems vs. standard HPO problems.
* Unanimous decision of the reviewers (and confidently so).

**Justification For Why Not Lower Score:**

N/A

**Metareview: Summary, Strengths And Weaknesses:**

The reviewers and meta reviewer all carefully checked the rebuttal. They thank the authors for their response and their efforts during the rebuttal phase. While the reviewers and meta reviewer all acknowledge that it is a promising direction to investigate the HPO problem for federated learning, they have found the following concerns that remained after the rebuttal:

* Further develop the empirical evaluation in order to fully demonstrate the presented set of desiderata, for instance, the extensibility.
* Some reviewers continued to believe that the paper should further clarify/motivate in what sense the federated HPO problem differs from the standard HPO problem and why the existing HPO toolkit is not appropriate (though the revised manuscript makes a good step in that direction).

The reviewers and the meta reviewer believe that the accumulation of those flaws—one of them being quite substantial, implying experimental work—cannot be addressed within the reviewing process of ICLR 2023. Addressing those comments warrants another full cycle of peer reviews. As a result, the paper is recommended for rejection. The detailed comments of the reviewers provide an actionable list of items to improve the paper for a future resubmission.